# LoRA-XS: Low-Rank Adaptation with Extremely Small Number of Parameters

## Abstract

The rapid expansion of large language models (LLMs) has underscored the need for parameter-efficient fine-tuning methods, with LoRA (Low-Rank Adaptation) emerging as a popular solution. Although LoRA reduces the number of trainable parameters, serving multiple (task or user-specific) LoRA modules on top of a base model still creates significant storage challenges. To address this, using theoretical derivation, we introduce LoRA-XS (**Lo**w-**R**ank **A**daptation with e**X**tremely **S**mall number of parameters), a novel low-rank adaptation method that considerably reduces the trainable parameters while showing superior or competitive performance. LoRA-XS achieves this by inserting a small, trainable $r \times r$ weight matrix between frozen low-rank matrices, which are constructed by Singular Value Decomposition (SVD) of the original weight matrix. This lightweight matrix enables fine-tuning with drastically reduced storage requirements, making it feasible to deploy millions of personalized models while minimizing memory overhead. For instance, LoRA-XS achieves a remarkable reduction of trainable parameters by over 100x in 7B models compared to LoRA. Our evaluations across various benchmarks (including GLUE, GSM8K, MATH, and eight commonsense reasoning datasets) demonstrate that LoRA-XS performs competitively or better than LoRA and other recent methods like VeRA while being significantly more parameter efficient. We also provide an extensive ablation study on the importance of singular vectors in transformer weights, shedding light on the underlying mechanisms driving LoRA-XS's enhanced efficiency. These findings suggest that LoRA-XS is not only a storage-efficient alternative, but also a powerful tool for scaling and personalizing LLMs at unprecedented scales.[1]

## 1 Introduction

In recent years, the development of large language models (LLM) has revolutionized the field of natural language processing (NLP), enabling unprecedented performance across various tasks. However, these state-of-the-art models often come with a huge number of parameters, presenting significant challenges for fine-tuning and adaptation to specific downstream tasks. Modifying and storing these immense models introduces computational and storage challenges.

Given these challenges, Parameter-Efficient Fine-Tuning (PEFT) methods, where only a relatively small number of parameters are fine-tuned, emerged as a potential solution to compensate for the tremendous compute/storage cost of full parameter fine-tuning (Houlsby et al., 2019; Hu et al., 2021; Lester et al., 2021; Li & Liang, 2021; Zaken et al., 2021). Among PEFT methods, LoRA (Hu et al., 2021) is widely used in recent literature due to its good generalization and the fact that it does not introduce extra modules during the inference phase. Nevertheless, even these techniques can require considerable storage and computational resources, particularly when the objective is to enable large-scale personalized or task-specific adaptation. As an example, applying LoRA on the GPT-3 model (Brown et al., 2020) with a rank of 16 and only query and value matrices being adapted, would result in 144MB of memory per checkpoint, which would amount to 144TB of memory when serving 1 million personalized models.

Following LoRA, many successors have been proposed to further reduce the number of parameters and improve efficiency (Kopiczko et al., 2023; Liu et al., 2024; Zhang et al., 2023). One such

---

[1]We will release code upon acceptance.

recent state-of-the-art method is VeRA (Kopiczko et al., 2023), which reduces the number of trainable parameters by freezing the LoRA matrices and using a single pair of low-rank matrices shared across all layers while learning small scaling vectors instead. Although VeRA improves parameter efficiency, its parameter count remains dependent on the model hidden dimensions, which becomes increasingly significant for larger language models.[2] This dependency can result in substantial storage and computational requirements as model sizes continue to grow.

In this paper, by applying theoretical justification, we propose LoRA-XS, a highly parameter-efficient LoRA-based method. LoRA-XS is designed to achieve similar or superior adaptation performance with significantly fewer parameters – in particular, the number of LoRA-XS trainable parameters is independent of the model's hidden dimensions. Consequently, LoRA-XS marks a new paradigm in parameter-efficient fine-tuning (PEFT), overcoming the limitations of existing approaches and offering a more efficient path to model personalization and task-specific optimization. Using the previous example on low-rank adaptation of GPT-3 with a rank of 16, serving 1 million personalized models with LoRA-XS would require only 96GB of storage, compared to LoRA's 144TB, resulting in more than a 1500x reduction in storage requirements.

Moreover, with LoRA-XS, we can precisely control the number of additional parameters, allowing for flexible memory usage (see Figure 1). This flexibility is particularly beneficial for increasingly larger models, where traditional methods impose a certain minimum number of additional parameters. Furthermore, LoRA-XS retains the core advantages of LoRA, such as not requiring any modifications to the model architecture and introducing no additional latency during inference, making it an efficient and seamless solution for practical deployment.

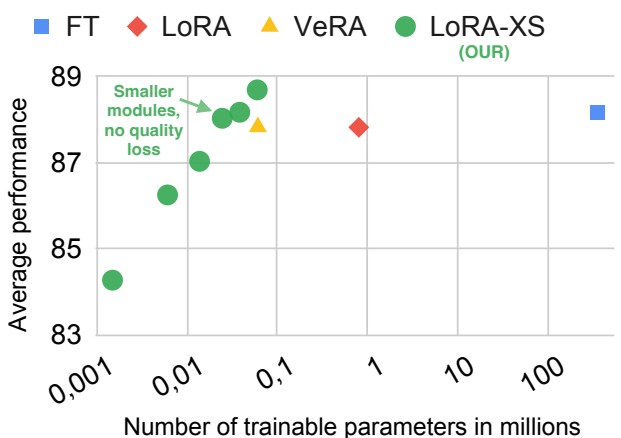

Figure 1: Average performance of RoBERTa-large on a subset of GLUE tasks (see Table 1) as a function of the number of trainable parameters (in millions) for different adaptation methods: Full Fine-Tuning (FT), LoRA, VERA, and LoRA-XS. LoRA-XS points correspond to various ranks from 4 up to 25. LoRA-XS consistently outperforms other methods in both average performance and parameter efficiency. Unlike other approaches, LoRA-XS provides greater flexibility in reducing the number of trainable parameters, as it is not constrained by model dimension, enabling more efficient adaptation without a lower bound.

LoRA-XS achieves this extreme parameter efficiency by setting LoRA's projection matrices using Singular Value Decomposition (SVD) of the pre-trained module weights and keeping them frozen during training (see Figure 2). The only trainable parameter of LoRA-XS is an $r \times r$ matrix (*i.e.*, $R$) between the frozen LoRA projection matrices, where $r$ denotes LoRA's rank. Fixing these matrices during training transforms our method into a *latent editing* approach, with the matrix $R$ using only $r^2$ parameters. Although LoRA-XS trains the model in a constrained parameter space, as we will show in later sections, its performance remains competitive or better than the LoRA baseline and more recent methods like VeRA across various benchmarks and model scales. We demonstrate LoRA-XS's performance across a wide range of benchmarks, including GLUE (Wang et al., 2018) for natural language understanding, GSM8K (Cobbe et al., 2021) and MATH (Hendrycks et al., 2021) for mathematical reasoning, and eight commonsense reasoning datasets (see Section 4).

---

[2]For example, while early transformer models like BERT have a hidden dimension of 768, the recent GPT-3 model has a hidden dimension of 12288, which directly affects the trainable parameter count for LoRA and VeRA methods.

We also conduct an extensive ablation study, revealing the essential role of singular vectors in transformer weights, which highlights the core mechanism behind LoRA-XS's efficiency (see Section 5). Our findings suggest that LoRA-XS is not only highly parameter-efficient but also a powerful enabler for scaling and personalizing large language models at unprecedented scales.

In summary, our contributions are as follows:

- By applying theoretical derivation, we introduce LoRA-XS, a highly parameter-efficient fine-tuning method that reduces the number of trainable parameters by over 100x in large-scale models without compromising performance.

- LoRA-XS outperforms LoRA and other recent approaches such as VeRA across various model sizes (including 7B and 8B LLMs) and a broad range of tasks, including GLUE, GSM8k, MATH, and eight commonsense reasoning benchmarks.

- Unlike existing LoRA variants, LoRA-XS offers unprecedented flexibility, with its parameter count being independent of model dimensions, making it more storage-friendly and adaptable (see Figure 1).

## 2 RELATED WORK

**Efficient Adaptation**    Recently, there has been many variants of adapter-based fine-tuning methods proposed where a set of adapter modules are introduced into the transformer model (Vaswani et al., 2017). These modules can either be introduced as extra *adapter* layers into the transformer block (Houlsby et al., 2019; Liu et al., 2022; Pfeiffer et al., 2020), or as an additional set of parameters modifying input layer activations (Asai et al., 2022; Lester et al., 2021; Li & Liang, 2021; Liu et al., 2023). Although these approaches introduce a relatively small number of parameters, they deteriorate model's latency in online inference, especially in large-scale production scenarios.

**Low-rank Adaptation**    Low-rank adaptation of transformer models, proposed by LoRA (Hu et al., 2021), offers a strong alternative to previous PEFT methods, where the generalization performance is competitive to full fine-tuning while not introducing any further latency during inference phase. Building upon the LoRA method, there has been many recent efforts to improve its learning curve (Liu et al., 2024; Hayou et al., 2024; Meng et al., 2024), reduce the trainable parameters (Zhang et al., 2023; Kopiczko et al., 2023; Renduchintala et al., 2023), or even training it with quantized pre-trained weights to improve memory footprint during training (Dettmers et al., 2024; Li et al., 2023). Our method, LoRA-XS, falls in the second category, where we aim to significantly reduce trainable parameters while performing competitively to LoRA over various benchmarks and different model scales.

**Parameter-constrained LoRA variants**    Several recent works propose variants of LoRA that reduce the number of trainable parameters while maintaining competitive performance. AdaLoRA (Zhang et al., 2023) investigates a dynamic rank adjustment for different modules' low-rank matrices as opposed to uniform parameter allocation in LoRA. Tied-LoRA (Renduchintala et al., 2023) improves the parameter efficiency by tying LoRA matrices across all layers of the transformer model. VeRA (Kopiczko et al., 2023), which is closely related to our work, shares randomly initialized frozen LoRA matrices across layers and adds trainable scaling vectors. However, unlike VeRA, LoRA-XS initializes its low-rank matrices using the SVD of the pre-trained model weights, providing both theoretical justification (see Section 3.1) and strong empirical performance (see Section 5). Additionally, LoRA-XS's number of trainable parameters is independent of the model's hidden dimensions, allowing for a significant reduction in parameters, particularly in large-scale models.

## 3 METHOD

This section introduces LoRA-XS (**Lo**w-**R**ank **A**daptation with e**X**tremely **S**mall parameters), a novel method designed to improve the parameter efficiency of fine-tuning large language models by leveraging insights from low-rank adaptation. Building on LoRA's core ideas, our approach addresses scalability and storage issues while maintaining highly competitive performance.

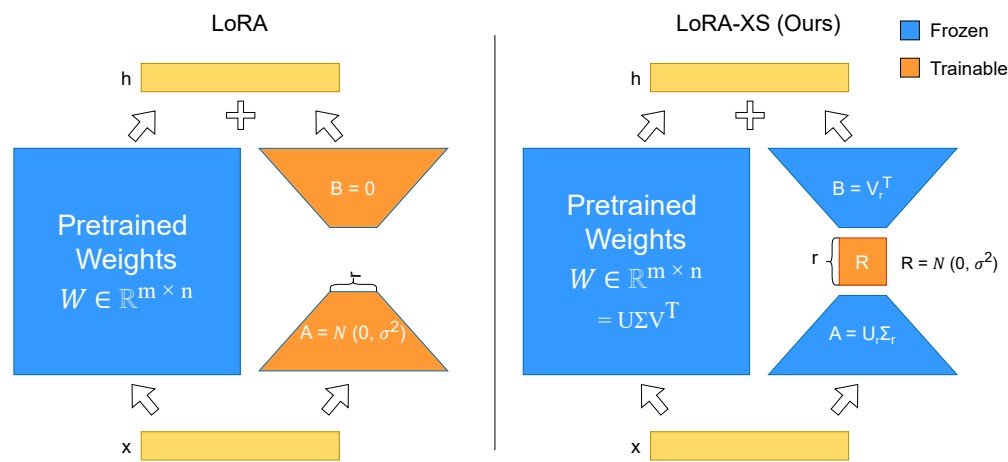

Figure 2: A visual comparison of the LoRA and LoRA-XS techniques. The key innovation of LoRA-XS is its use of a small trainable matrix $R$ positioned between two frozen low-rank matrices, $A$ and $B$, which are derived from the truncated SVD of pre-trained weights, retaining the top $r$ singular components. LoRA-XS allows for extreme parameter efficiency while maintaining performance.

In recent years, LoRA (Hu et al., 2021) has been pivotal in parameter-efficient tuning by introducing low-rank matrices for adaptation, significantly lowering the number of trainable parameters. However, deploying LoRA at scale, particularly across large-scale task-specific or user-specific models, can substantially increase the required storage needs. As modern models grow in size and complexity, the need for even more parameter-efficient tuning strategies becomes a crucial issue.

LoRA demonstrated that its low-rank weight updates (*i.e.*, $\Delta W$) align with certain *directions* already present in the model's weights. Building on this insight and our theoretical derivation (see Section 3.1), we propose initializing the LoRA adaptation matrices ($A$ and $B$ in Figure 2) using the top singular vectors from the SVD of the pre-trained weight matrix $W$. These matrices are kept fixed during training. To introduce flexibility, we add a trainable $r \times r$ matrix $R$ between $A$ and $B$, making $R$ the only learnable component. This drastically reduces the number of trainable parameters, with the parameter count independent of model dimensions. Figure 2 provides an overview of the LoRA-XS method, highlighting its differences from the original LoRA framework.

## 3.1 THEORETICAL DERIVATION OF LoRA-XS

In this section, we derive the theoretical foundations of our approach, with full details provided in Appendix A. Readers primarily interested in the practical aspects of LoRA-XS can proceed to the subsequent sections.

We begin by considering a neural network with transformer architecture. Let $W \in \mathbb{R}^{n \times n}$ be a square weight matrix for an arbitrary linear layer in this network. Our goal is to adapt the weights to new tasks by applying a correction $\Delta W$, which we want to constrain to a lower-dimensional subspace of $\mathbb{R}^{n \times n}$. We aim to show how to choose such a subspace to allow for significant flexibility and to work effectively for future gradient adaptations.

In the standard LoRA method, the subspace used for adaptation is characterized as the set of all matrices of the form $AB$, where $A \in \mathbb{R}^{r \times n}$ and $B \in \mathbb{R}^{n \times r}$, resulting in a space with dimension equal to $2nr$. However, in our proposed LoRA-XS framework, we introduce a more general parametrization of subspaces, which allows for a dimension $r^2$, where $r$ can range from 1 to $n$. Compared to LoRA, which has at least dimension of $2n$, our approach can use an arbitrarily small amount of memory (see Figure 1).

Formally, given fixed orthogonal matrices $A \in \mathbb{R}^{r \times n}$ and $B \in \mathbb{R}^{n \times r}$, we define the subspace $S_{A,B}^r$ as

$$S_{A,B}^r = \{AXB^T : X \in \mathbb{R}^{r \times r}\}.$$

This subspace has dimension $r^2$ and allows for a flexible choice of $r$ to adjust the dimensionality. Moreover, we can easily compute the orthogonal projection onto $S_{A,B}^r$. Namely,

$$p_{A,B}(X) = A[A^T X B]B^T \text{ for } X \in \mathbb{R}^{n \times n},$$

is the orthogonal projection with respect to Frobenius scalar product in the space of matrices on $S_{A,B}^r$ (the proof of this result is presented in Appendix A). This projection can be useful when we want to project a full gradient onto the space $S_{A,B}^r$.

**Main idea behind LoRA-XS** The problem that LoRA-XS aims to solve is how to choose the matrices $A$ and $B$ such that the following optimization procedures yield similar weights:

- fine-tuning the network's weights without any restrictions,
- fine-tuning the network's weights restricted to $S_{A,B}^r$.

We demonstrate that, under reasonable assumptions, the optimal matrices $A$ and $B$ are obtained through truncated SVD on the initial weights $W$.

Consider the fine-tuning of the model. Assume we have a pre-trained weight matrix $W \in \mathbb{R}^{n \times n}$, and that we want to find the space $S_{A,B}^r$ in which the modification would be optimal for the further fine-tuning. Let $G_1, \ldots, G_k$ denote the gradients computed for the mini-batches. During the SGD optimization with learning rate $h$, we would arrive at the weights:

$$W + \Delta W = W + hG_1 + \ldots + hG_k.$$

If we have chosen our subspace $S_{A,B}^r$ such that $\Delta W$ is close to it, we can transition from $S_{A,B}^r$ to $\Delta W$ during fine-tuning. More precisely, if $\Delta W \in S_{A,B}^r$, then $p_{A,B}(\Delta W) = \Delta W$, and by applying the orthogonal projection in our model, we get:

$$W + hp_{A,B}(G_1) + \ldots + hp_{A,B}(G_k) = W + p_{A,B}(hG_1 + \ldots + hG_k) = W + p_{A,B}(\Delta W) = W + \Delta W.$$

**Theorem 3.1.** *Let $G$ denote the mean gradient: $G = \frac{1}{k}\sum_{i=1}^k G_i$. Let us apply truncated SVD decomposition on $G$ to obtain $U_r, \Sigma_r, V_r$. Then*

$$U_r, V_r = \underset{A,B}{\arg\min}\, d(G; S_{A,B}^r),$$

*where $d$ denotes the distance in the Frobenius norm.*

*Proof.* Observe that every element of $S_{A,B}^r$ is trivially of rank at most $r$. By the matrix approximation lemma, known as the Eckart–Young–Mirsky theorem, we know that the optimal approximation in Frobenius norm of matrix $G$ in the rank $r$ matrices is given by $U_r, \Sigma_r, V_r^T$, where $G = U\Sigma V^T$ is the SVD decomposition of $G$. $\square$

To extend this result to LoRA-XS, we make the additional assumption that the gradients during fine-tuning do not essentially diverge from those observed during pre-training. In practice, this is often the case, as fine-tuning tasks are typically similar to pre-training tasks, resulting in a shift in the distribution of gradients rather than an entirely new distribution. Indeed, the efficiency of LoRA-XS, as demonstrated in our experiments, supports this assumption (see Section 4). Moreover, our ablation study on LoRA-XS initialization shows a significant benefit when using the SVD of the weights for tasks aligned with language modeling. For SST-2, this initialization provided similar accuracy to random initialization (see Table 4 and Appendix F), which may stem from the fact that this task is not well aligned with language modeling compared to the other tested tasks, such as CoLA, MRPC, and QNLI.

*Observation* 3.1. Consider a deep neural network $\Phi$ pre-trained on a dataset $x_1, \ldots, x_k$, with weights $W$. Let $G_i = \nabla\Phi(x_i)$ be the gradients of the pre-trained model. Then,

$$W \sim \frac{1}{k}\sum_{i=1}^k G_i.$$

This follows from the fact that at the final phase of the neural networks training, its weights stabilize. Consequently, at the final phase of training, the previous gradients $\tilde{G}_t$ computed at iteration $t$ are

close to the ones computed for the fully trained model. Then for $W_t$ denoting the weights of the model in the step $t$, we get:

$$W_T = W_t + h\tilde{G}_t + \ldots + h\tilde{G}_{T-1} \approx W_t + hG_t + \ldots + hG_{T-1}.$$

Consequently, the gradients accumulate and represent the largest part of the sum:

$$W_T \approx h(G_t + \ldots + G_{T-1}).$$

Since gradients come from the same distribution we obtain $W = W_T \sim \frac{1}{k}\sum_i G_i$.

This leads us to the formulation of LoRA-XS. Suppose we aim to obtain a good $d$-dimensional approximation for the optimal gradient training of a linear layer with weights $W$ in a pre-trained transformer model, where $d = r^2$. To do this, we apply truncated (to $r$ eigenvalues) SVD to weights $W$ obtaining $U_r, \Sigma_r, V_r^T$. Then, to make an update with learning rate $h > 0$:

- compute the gradient $G_i$ for the model over the consecutive $i$-th mini-batch,
- update the weights by taking the projection of the computed gradient on the space $S_{U_r,V_r}^r$:

$$\Delta W_i = h \cdot U_r[U_r^T G_i V_r]V_r^T.$$

In practice, the projection is not necessary, as we can work directly in the space $S_{U,V}^r$. Thus, we can compute the gradient with respect to $r \times r$ dimensional update space:

$$W + U_r R V_r^T \text{ for trainable } R \in \mathbb{R}^{r \times r}.$$

Observe that, due to the truncated SVD decomposition, the matrix $R$ has a size of $r \times r$. The next subsection provides a more direct description of our method. Additionally, please refer to the experiments and ablation sections (see Section 4 and Section 5), which empirically support our theory of LoRA-XS. One additional factor we observed empirically is that it is beneficial to rescale $R$ by the components of $\Sigma_r$ (see Section 5 and Appendix H). Thus, in LoRA-XS, we optimize in the space $W + U_r \Sigma_r R V_r^T$, where $R$ is the trainable matrix with $r \times r$ coefficients.

## 3.2 FORMULATION OF LoRA-XS

The main idea behind the LoRA-XS is to modify the adaptation process by introducing a small square matrix $R \in \mathbb{R}^{r \times r}$ between frozen LoRA matrices that are set using truncated SVD of the pre-trained weight matrix $W \in \mathbb{R}^{m \times n}$. [3]

The traditional LoRA forward path for an input $x \in \mathbb{R}^n$ can be formulated as:

$$h = xW + x\,\Delta W = xW + xAB$$

where $\Delta W \in \mathbb{R}^{m \times n}$ is the low-rank weight update. The matrices $A \in \mathbb{R}^{m \times r}$ and $B \in \mathbb{R}^{r \times n}$ are low-rank matrices with $r \ll \min(m,n)$. During training, $W$ is kept frozen, and $A$ and $B$ are the trainable parameters.

In LoRA-XS, we improve parameter efficiency by introducing a small trainable matrix $R \in \mathbb{R}^{r \times r}$, while keeping matrices $A$ and $B$ frozen, modifying the forward path to:

$$h = xW + x\,\Delta W = xW + xARB$$

Here, $A$ and $B$ are set using the truncated SVD of the original weight matrix $W$. Formally, the SVD of $W$ is given by:

$$W = U\Sigma V^T$$

where $U \in \mathbb{R}^{m \times m}$, $\Sigma \in \mathbb{R}^{m \times n}$, and $V \in \mathbb{R}^{n \times n}$. We set (frozen) matrices $A$ and $B$ as:

$$A = U_r \Sigma_r \quad \text{and} \quad B = V_r^T$$

where $U_r \in \mathbb{R}^{m \times r}$ and $V_r \in \mathbb{R}^{n \times r}$ contain the left/right singular vectors corresponding to the top $r$ singular values, and $\Sigma_r \in \mathbb{R}^{r \times r}$ is a diagonal matrix that contain these top $r$ singular values of $\Sigma$.

---

[3]From now on, to maintain consistency with common notation conventions and our LoRA-XS code, we will work in the transposed space, where vectors are represented as rows, and the multiplication of a vector $x$ by a matrix $A$ is expressed as $xA$. Consequently, $W$ will formally denote the transposed weight matrix.

Our newly introduced $R$ matrix is initialized with a Gaussian distribution $N(0, \sigma^2)$, where $\sigma$ is set to a small but non-zero value[4]. This ensures that we start fine-tuning with a model almost identical to the original pre-trained model. During fine-tuning, the matrices $A$ and $B$ are kept frozen, and only $R$ is updated, significantly reducing the number of trainable parameters.

Compared to the other LoRA variants, LoRA-XS provides better control over the number of trainable parameters, allowing for more flexible required storage for the fine-tuned models. This flexibility is particularly beneficial for increasingly larger models, where traditional methods are often limited by model's hidden dimensions. Similar to LoRA and its successors, LoRA-XS does not introduce any extra computational overhead or latency during inference, as this module can be merged into the original matrix post-training.

We make the following observation on the parameter efficiency of LoRA-XS compared to LoRA and VeRA methods (please refer to Appendix B for the details).

**Observation:** LoRA-XS demonstrates superior parameter efficiency compared to both LoRA and VeRA.

Let's consider a transformer model with $L$ fine-tuned layers, each consisting of $q$ number of $W \in \mathbb{R}^{n \times n}$ matrices. As the model dimension $n$ becomes very large compared to the rank $r$, the benefit of LoRA-XS over LoRA and VeRA becomes more pronounced. Specifically for large $n$:

$$\frac{P_{\text{LoRA}}}{P_{\text{LoRA-XS}}} \approx \frac{2n}{r} \quad \text{and} \quad \frac{P_{\text{VeRA}}}{P_{\text{LoRA-XS}}} \approx \frac{n}{r^2}.$$

This indicates that for large models, LoRA and VeRA require significantly more parameters than LoRA-XS, with the difference growing linearly with $n$ (*i.e.*, model's hidden dimension). This makes LoRA-XS especially suitable for fine-tuning LLMs models where parameter efficiency is crucial.

## 4 EXPERIMENTS

This section describes experiments evaluating the effectiveness of LoRA-XS. We begin by detailing the experimental setup and then then provide results for the GLUE benchmark (Wang et al., 2018), where we compare LoRA-XS with full fine-tuning, LoRA, and VeRA across six tasks. We explore various ranks for LoRA-XS to highlight their effect on performance and parameter efficiency.

We also report results on instruction tuning experiments with decoder-only language models. These experiments test LoRA-XS's ability to enable large language models (LLMs) to follow instructions with minimal parameter overhead. The first set of experiments involves training models on the MetaMathQA dataset (Yu et al., 2023) and evaluating them on the GSM8K (Cobbe et al., 2021) and MATH (Hendrycks et al., 2021) benchmarks, focusing on mathematical reasoning. The second set evaluates commonsense reasoning in the same setting as Hu et al. (2023), using eight benchmarks to assess model performance.

Our preliminary findings show that when the number of trainable parameters is limited, using a lower rank $r$ with more LoRA-XS modules yields better results. This guided our strategy of using smaller ranks while distributing more LoRA-XS modules than LoRA and VeRA, which mainly added adaptation modules to the Query and Value matrices in the main experiments. By spreading LoRA-XS across additional components and keeping the rank low, we achieve a balanced parameter allocation without significantly increasing the number of trainable parameters.

### 4.1 EXPERIMENTAL SETUP

For the GLUE benchmark experiments, we use the RoBERTa-large model (Liu et al., 2019) and explore different ranks for LoRA-XS, ranging from $r = 4$ to $r = 25$. This range allows us to examine the impact of varying numbers of trainable parameters on performance. For GLUE experiments, we add LoRA-XS modules to the Query, Value, Attention Output, and the Output Fully Connected weight matrices in all layers of the RoBERTa model. Hyperparameters were selected through grid search, and the chosen values are summarized in Table 5. Similar to previous efforts, when fine-

---

[4]In all LoRA-XS experiments, we set $\sigma$ to $10^{-5}$.

tuning the model on MRPC, RTE and STS-B tasks, the model is initialized using weights fine-tuned on the MNLI task[5] (Hu et al., 2021).

For the mathematical reasoning experiments, we use the Mistral-7B-v0.1 (Jiang et al., 2023) and Gemma-7B (Team et al., 2024) decoder-only models, training them on 100k samples of the MetaMathQA (Yu et al., 2023) dataset and evaluating on the GSM8K (Cobbe et al., 2021) and MATH (Hendrycks et al., 2021) datasets.

For the commonsense reasoning experiments, we use LLaMA2 7B (Touvron et al., 2023) and LLaMA3 8B (Dubey et al., 2024) decoder-only models, fine-tune them on a mixture of eight sub-tasks[6], and then separately evaluate the fine-tuned models on the validation set of these eight datasets (see section C.2 for more details). Our training/evaluation setting follows prior work (Hu et al., 2023) in order to have a fair comparison with LoRA as the baseline method.

For all instruction tuning experiments, LoRA-XS modules are added to the Query, Key, Value, Attention Output, and all three fully connected weight matrices. Each LoRA-XS module in our main experiments is initialized with the SVD of the corresponding pre-trained weights $W$. Further details of the experimental setup are provided in the Appendix C.

## 4.2 GLUE BENCHMARK

In Table 1, we present the performance of the RoBERTa-large model on the GLUE benchmark using full fine-tuning (FT) and parameter-efficient fine-tuning methods: LoRA, VeRA and our LoRA-XS.

| Method | # Trainable Parameters | Rank | SST-2 | MRPC | CoLA | QNLI | RTE | STS-B | Avg. |
|---|---|---|---|---|---|---|---|---|---|
| FT | 355,000K | - | 96.4 | 90.9 | 68.0 | 94.7 | 86.6 | 92.4 | 88.17 |
| LoRA | 800K | 8 | $96.2 \pm 0.5$ | $90.2 \pm 1.0$ | $68.2 \pm 1.9$ | $94.8 \pm 0.3$ | $85.2 \pm 1.1$ | $92.3 \pm 0.5$ | 87.82 |
| VeRA | 61K | 256 | $96.1 \pm 0.1$ | $90.9 \pm 0.7$ | $68.0 \pm 0.8$ | $94.4 \pm 0.2$ | $85.9 \pm 0.7$ | $91.7 \pm 0.8$ | 87.83 |
| LoRA-XS | 60K | 25 | $96.33 \pm 0.15$ | $91.18 \pm 0.82$ | $68.55 \pm 0.81$ | $94.34 \pm 0.22$ | $89.53 \pm 0.48$ | $92.22 \pm 0.10$ | 88.69 |
| | 38.4K | 20 | $95.87 \pm 0.28$ | $90.44 \pm 0.41$ | $68.08 \pm 1.21$ | $94.05 \pm 0.16$ | $88.81 \pm 0.20$ | $91.76 \pm 0.18$ | 88.17 |
| | 24.6K | 16 | $95.87 \pm 0.24$ | $90.69 \pm 0.37$ | $66.96 \pm 1.23$ | $93.89 \pm 0.06$ | $88.81 \pm 0.30$ | $91.98 \pm 0.13$ | 88.03 |
| | 13.8K | 12 | $95.87 \pm 0.31$ | $90.20 \pm 0.32$ | $65.47 \pm 0.90$ | $93.32 \pm 0.51$ | $87.73 \pm 0.68$ | $91.40 \pm 0.12$ | 87.03 |
| | 6.1K | 8 | $95.30 \pm 0.34$ | $88.48 \pm 0.64$ | $64.39 \pm 0.75$ | $92.49 \pm 0.09$ | $86.28 \pm 0.59$ | $90.57 \pm 0.25$ | 86.25 |
| | 1.5K | 4 | $94.84 \pm 0.29$ | $87.75 \pm 0.33$ | $60.52 \pm 1.54$ | $90.94 \pm 0.27$ | $82.67 \pm 0.53$ | $88.88 \pm 0.22$ | 84.27 |

Table 1: RoBERTa-large performance on a selection of tasks from the GLUE benchmark with different adaptation methods. We report Matthew's correlation for CoLA, Pearson correlation for STS-B, and accuracy for the other tasks. Full fine-tuning, LoRA, and VeRA results are taken from prior works (Hu et al., 2021; Kopiczko et al., 2023). Results indicate the median and standard deviation of five runs with different seeds. LoRA-XS demonstrates competitive or superior performance over the baselines while offering much better parameter efficiency. Higher is better for all metrics.

As shown in Table 1, LoRA-XS with ranks of 25, 20, and 16 (corresponding to 60K, 38.4K, and 24.6K trainable parameters, respectively) outperform baseline PEFT methods (LoRA and VeRA), achieving the highest average performance across the tested GLUE tasks. Notably, with a rank of 16, LoRA-XS achieves better accuracy than VeRA while having 2.5x less trainable parameters. Following our baselines' evaluation method, we conduct 5 runs with different seeds, recording the best epoch's outcome for each run, and report the median of these results. Similar to LoRA (Hu et al., 2021) and VeRA (Kopiczko et al., 2023), we only include the added LoRA-XS modules in the calculation of trainable parameters count, excluding the classifier parameters for a clear comparison.

We observe competitive performance even at extremely low ranks, highlighting LoRA-XS's parameter efficiency. While performance drops slightly as the rank decreases, this is expected due to the reduced parameter count. The most notable decline occurs at the smallest rank of 4, with an accuracy drop of about 4 percentage points. Despite using only around 1500 parameters, LoRA-XS retains strong performance, demonstrating its ability to maintain competitive results while drastically reducing trainable parameters.

## 4.3 INSTRUCTION TUNING

---

[5]The model is trained for 10 epochs on the MNLI dataset and then further fine-tuned on these three tasks.

[6]The training dataset is a collection of 170K commonsense reasoning samples derived from Hu et al. (2023).

| Model | Method | # Trainable Parameters | BoolQ | PIQA | SIQA | HellaSwag | WinoGrande | ARC-e | ARC-c | OBQA | Avg. |
|---|---|---|---|---|---|---|---|---|---|---|---|
| LLaMA2-7B | LoRA | 56M | 69.8 | 79.9 | 79.5 | 83.6 | 82.6 | 79.8 | 64.7 | 81.0 | 77.6 |
| | LoRA-XS | 0.23M | 67.2 | 81.8 | 78.1 | 75.4 | 80.8 | 81.2 | 65.9 | 74.6 | 75.6 |
| LLaMA3-8B | LoRA | 57M | 70.8 | 85.2 | 79.9 | 91.7 | 84.3 | 84.2 | 71.2 | 79.0 | 80.8 |
| | LoRA-XS | 0.23M | 66.6 | 85.8 | 79.4 | 90.1 | 85.2 | 87.0 | 76.5 | 81.8 | 81.6 |

Table 2: Accuracy evaluation of fine-tuned LLaMA2 7B and LLaMA3 8B models with LoRA and LoRA-XS methods across eight commonsense reasoning datasets. The rank is set to 32 in all evaluation settings. The LoRA results are taken from prior work (Liu et al., 2024). LoRA-XS outperforms LoRA for the LLaMA3-8B model and is competitive with LoRA for the LLaMA2-7B model, while using only ∼0.4% of LoRA's trainable parameters.

The results of our instruction tuning experiments are presented in Table 2 and Table 3.

Table 2 compares LoRA-XS performance with LoRA over eight commonsense reasoning datasets. We can observe that for both LLaMA2 and LLaMA3 models, LoRA-XS performs competitively or better than LoRA while having only ∼0.4% of LoRA's trainable parameters at the same *rank*.

For mathematical reasoning, as demonstrated in Table 3, applying LoRA-XS to a 7B-scale model performs competitively with both LoRA and full fine-tuning, showcasing the applicability of our method to larger-scale models. Notably, LoRA-XS with only 0.92M parameters (rank=64) achieves close performance to LoRA, which uses 168M parameters, across both the GSM8k and MATH benchmarks. This represents a reduction of over 150x in the number of trainable parameters.

| Model | Method | Rank | # Trainable Parameters | GSM8K | MATH |
|---|---|---|---|---|---|
| Mistral (7B) | Full FT | - | 7242M | 67.02 | 18.60 |
| | LoRA | 64 | 168M | 67.70 | 19.68 |
| | LoRA-XS | 64 | 0.92M | 68.01 | 17.86 |
| | | 32 | 0.23M | 63.23 | 15.88 |
| | | 16 | 0.057M | 57.92 | 14.44 |
| Gemma (7B) | Full FT | - | 8538M | 71.34 | 22.74 |
| | LoRA | 64 | 200M | 74.90 | 31.28 |
| | LoRA-XS | 64 | 0.80M | 74.22 | 27.62 |
| | | 32 | 0.20M | 71.72 | 27.32 |
| | | 16 | 0.050M | 68.46 | 26.38 |

Table 3: Instruction tuning performance on GSM8K and MATH Benchmarks for Mistral-7B and Gemma-7B models using full fine-tuning, LoRA, and LoRA-XS. Full fine-tuning and LoRA performance values are taken from prior work (Meng et al., 2024). Higher is better for all metrics. LoRA-XS performs competitively or better than both LoRA and full fine-tuning while being significantly more parameter efficient in the studied settings.

## 5 ABLATION

In this section, we present ablation experiments to better understand the efficiency of LoRA-XS, demonstrate our theoretical derivations in practice, and examine the role of singular vectors in transformer weights and adaptation layers.

**Importance of Singular Vectors in Transformer Weights** We begin by analyzing the significance of singular vectors in the weight matrices of transformer models. Detailed results can be found in Appendix D. By examining different subsets of singular vectors (top, middle, and bottom) for various transformer weights (e.g., attention and feedforward modules), we find that the top singular vectors retain the most task-relevant knowledge. In contrast, the middle and bottom singular vectors contribute less to task performance, suggesting that they encode more subtle or less critical information. This observation aligns with our proposal to initialize LoRA-XS using top singular vectors.

**Delta Weight Approximation and Singular Subspace Retention** We evaluate how accurately the full weight update $\Delta W$, obtained during fine-tuning, can be approximated by projecting it onto different subspaces of singular vectors from the SVD of the original pre-trained weight matrix $W$. Specifically, we experiment with retaining varying fractions of top, middle, and bottom singular vectors and assess their impact on downstream task performance. As summarized in Appendix E, the self-attention modules (query, key, value, attention output) exhibit minimal performance degradation when only 1% or 10% of singular vectors are retained, regardless of whether they come from the top or bottom subspaces. In contrast, output dense layers are more sensitive to these approximations and

require a higher fraction of singular vectors to preserve accuracy. These results suggest that while self-attention layers can tolerate significant dimensionality reduction, output dense layers benefit from retaining a larger portion of the singular spectrum.

**LoRA-XS Initialization**   The initialization of matrices $A$ and $B$ is a key factor in the performance of LoRA-XS. We investigate three initialization strategies: random initialization, SVD of random matrices (SVD of random), and SVD of pre-trained weights (SVD of $W$). Please refer to Appendix F for the details.

As summarized in Table 4, our experiments show that using SVD on the pre-trained weight matrices generally leads to superior performance. This observation aligns with our theoretical framework, which claims that, assuming the considered task is similar to the task used for pre-training, SVD of the original weight matrix is the most effective initialization choice (see Section 3.1).

An exception to this trend is observed in the SST-2 task, where SVD of random matrices slightly outperforms SVD of $W$. We hypothesize that this is due to SST-2 being a sentiment classification task, which may not align as closely with the pre-training objective of language modeling as other tasks such as MRPC, CoLA, and QNLI. This insight reinforces our theoretical analysis, which suggests that SVD of pre-trained weights is most advantageous when the fine-tuning task shares similarities with the pre-training objective.

Additionally, we show that initializing LoRA-XS with SVD of the pretrained weights accelerates convergence in the early stages of LoRA-XS training (see Table 14). This early advantage sets LoRA-XS apart from other ultra-efficient adaptation techniques, such as soft prompt tuning (Lester et al., 2021; Li & Liang, 2021), which often exhibit slower convergence. By initializing $A$ and $B$ with information derived from the pre-trained model, LoRA-XS benefits from a more informed starting point, leading to more efficient and effective training.

**Top vs. Bottom Singular Vector Initialization**   In Appendix G, we further analyze whether it is more effective to initialize LoRA-XS with top or bottom singular vectors. Our analysis indicates that retaining the top singular vectors consistently yields better performance for LoRA-XS across various tasks.

**Including Singular Values in Initialization**   Lastly, in Appendix H, we evaluate whether including singular values $\Sigma$ in the initialization of matrix $A$ enhances the performance of LoRA-XS. The results indicate improved performance when $\Sigma$ is included in most cases, suggesting that while singular values do not alter the direction of the corresponding singular vectors, they may play a crucial role in scaling and emphasizing their significance. However, in one task, we observed better scores without $\Sigma$, which may suggest that certain types of tasks could benefit from a different approach to initialization.

| Init. Type | SST-2 | COLA | MRPC | QNLI |
|---|---|---|---|---|
| random | 94.72 | 58.53 | 85.78 | 88.80 |
| SVD of random | **94.84** | 55.27 | 84.31 | 88.34 |
| SVD of W | 94.72 | **60.11** | **87.50** | **90.94** |

Table 4: Performance of LoRA-XS with various initialization schemes. We present the best median scores across different learning rates, averaged over 5 seeds for rank 4. We report Matthew's correlation for CoLA and accuracy for the other tasks. Initializing LoRA-XS using the SVD of pre-trained weights (SVD of $W$) outperforms other methods across most tasks. Please refer to Appendix F for further details.

## 6   CONCLUSION

We introduce LoRA-XS, a novel parameter-efficient fine-tuning method that drastically reduces the number of trainable parameters while preserving or enhancing model performance, supported by solid theoretical foundations. LoRA-XS combines low-rank adaptation with singular value decomposition (SVD), aligning adaptation matrices with the principal components of pre-trained weights. Our experiments on GLUE, GSM8K, MATH, and eight commonsense reasoning datasets across multiple models demonstrate that LoRA-XS outperforms both LoRA and VeRA in parameter efficiency, while achieving competitive results across diverse tasks. This method provides a highly efficient approach to model adaptation with substantial parameter savings.

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

## A    THEORETICAL DERIVATION OF LoRA-XS: PROJECTION

Given fixed orthogonal matrices $A \in \mathbb{R}^{r \times n}, B \in \mathbb{R}^{n \times r}$, recall that

$$S_{A,B}^r = \{AXB^T : X \in \mathbb{R}^{r \times r}\}.$$

We show that one can easily compute orthogonal projection on $S_{A,B}^r$. Namely,

$$p_{A,B}(X) = A[A^T X B]B^T \text{ for } X \in \mathbb{R}^{n \times n},$$

is the orthogonal projection with respect to Frobenius scalar product in the space of matrices on $S_{A,B}^r$.

To prove the above, let us recall that $p$ is an orthogonal projection iff $p^2 = p$ and $p = p^T$. To check if it is projection onto space $S$ we have to additionally verify if $p(x) \in S$ for arbitrary $x$ and $p(x) = x$ for $x \in S$.

Let us first check that $p_{A,B}^2 = p_{A,B}$:

$$p_{A,B}^2(X) = A(A^T A)A^T X B(B^T B)B^T = AA^T XBB^T = p_{A,B}(X).$$

Now we check if $p_{A,B} = p_{A,B}^T$. Since for a self adjoint map $C$ we have $\langle Cx, y \rangle = \langle x, CY \rangle$, and $AA^T, BB^T$ are self-adjoint, we get

$$\langle p_{A,B}X, Y \rangle = \langle AA^T XBB^T, Y \rangle = \text{tr}(AA^T XBB^T)^T Y$$

$$= \text{tr}BB^T X^T AA^T Y = \text{tr}AA^T YBB^T X^T = \langle p_{A,B}Y, X \rangle.$$

Clearly, directly from definition $p_{A,B}(X) = A[A^T XB]B^T \in S_{A,B}^r$ for an arbitrary $X \in \mathbb{R}^{n \times n}$. Finally, we check that $p_{A,B}(W) = W$ for $W \in S_{A,B}^r$. Since $W \in S_{A,B}^r$, $W = AXB^T$ for some $X$. Consequently,

$$p_{A,B}(W) = AA^T(AXB^T)BB^T = AXB^T = W.$$

Thus we have show that the family $S_{A,B}^r$ of $r^2$-dimensional subspaces of $\mathbb{R}^{n \times n}$ allows an easy formula for orthogonal projection.

# B  PARAMETER EFFICIENCY OF LORA-XS

We make the following observation on the parameter efficiency of LoRA-XS compared to LoRA and VeRA methods.[7]

**Observation:** LoRA-XS demonstrates superior parameter efficiency compared to both LoRA and VeRA.

For simplicity, let's consider a transformer model with $L$ fine-tuned layers, each consisting of $q$ number of $W \in \mathbb{R}^{n \times n}$ matrices.

For LoRA, the number of trainable parameters is given by:

$$P_{\text{LoRA}} = L \times q \times r \times 2n, \tag{1}$$

For VeRA, the number of trainable parameters is given by:

$$P_{\text{VeRA}} = L \times q \times (n + r), \tag{2}$$

For LoRA-XS, the number of trainable parameters is given by:

$$P_{\text{LoRA-XS}} = L \times q \times r^2. \tag{3}$$

To compare the parameter efficiency, we compute the ratios of the number of trainable parameters between the methods. The ratio of trainable parameters for LoRA to LoRA-XS is:

$$\frac{P_{\text{LoRA}}}{P_{\text{LoRA-XS}}} = \frac{L \times q \times r \times 2n}{L \times q \times r^2} = \frac{2n}{r}, \tag{4}$$

Similarly, the ratio of trainable parameters for VeRA to LoRA-XS is:

$$\frac{P_{\text{VeRA}}}{P_{\text{LoRA-XS}}} = \frac{L \times q \times (n + r)}{L \times q \times r^2} = \frac{n + r}{r^2}. \tag{5}$$

As the model dimension $n$ becomes very large compared to the rank $r$, the benefit of LoRA-XS over LoRA and VeRA becomes more pronounced. Specifically for large $n$:

$$\frac{P_{\text{LoRA}}}{P_{\text{LoRA-XS}}} \approx \frac{2n}{r} \quad \text{and} \quad \frac{P_{\text{VeRA}}}{P_{\text{LoRA-XS}}} \approx \frac{n}{r^2}. \tag{6}$$

This indicates that for large models, LoRA and VeRA require significantly more parameters than LoRA-XS, with the difference growing linearly with $n$ (*i.e.*, model's hidden dimension). This makes LoRA-XS especially suitable for fine-tuning large language models where parameter efficiency is crucial.

To provide an example, for RoBERTa-large (Liu et al., 2019), which consists of 24 layers, assuming $q = 2$ (two additional trainable modules per layer), with each $W$ matrix of size $1024 \times 1024$ and $r = 16$, the number of trainable parameters for each method is as follows: For LoRA, $P_{\text{LoRA}} = 1,572,864$; for VeRA, $P_{\text{VeRA}} = 50,400$; for LoRA-XS, $P_{\text{LoRA-XS}} = 12,288$. Thus, LoRA requires about 31.2 times more parameters than VeRA and 128 times more than LoRA-XS, while VeRA requires about 4 times more parameters than LoRA-XS, highlighting the substantial parameter efficiency of LoRA-XS.

---

[7]It is worth noting that in these calculations we only consider the additional LoRA/VeRA/LoRA-XS parameters, and for encoder-only models, additional trainable parameters such as classifiers parameters may be added. However, these are common to all approaches and thus do not influence our comparative calculations.

## C EXPERIMENTAL SETUP DETAILS

In this section, we provide detailed information on the experimental setup and hyperparameters used in our experiments.

As mentioned in Section 4, for our main experiments, each LoRA-XS module is initialized using Singular Value Decomposition (SVD) of the corresponding pre-trained weight matrix $W$. The initialization process involves using truncated SVD (Halko et al., 2011). We take LoRA and VeRA scores from their respective papers (Hu et al., 2021; Kopiczko et al., 2023).

For all our experiments and the ablation study, models are trained with the AdamW optimizer (Loshchilov & Hutter, 2017), following methodologies from LoRA and VERA (Hu et al., 2021; Kopiczko et al., 2023). We utilize the HuggingFace Transformers library (Wolf et al., 2019) for Transformer-based models (Vaswani et al., 2017) and implement our LoRA-XS method on top of the Huggingface PEFT repository (Mangrulkar et al., 2022). Detailed setups and hyperparameters for each experiment are outlined in the following subsections.

### C.1 GLUE BENCHMARK

For the GLUE Benchmark experiments, we integrate LoRA-XS modules into the Query ($W_q$), Value ($W_v$), Attention Output ($W_o$), and first Fully Connected ($FC_1$) weight matrices of the transformer model (Vaswani et al., 2017). RoBERTa-large (Liu et al., 2019) serves as the base model. LoRA-XS's rank, $r$, is varied between 4 and 25, corresponding to 16 to 625 trainable parameters per module. Following previous work (Hu et al., 2021), LoRA modules for MRPC, RTE, and STS-B are initialized with weights fine-tuned on the MNLI task. Due to computational constraints, larger datasets like MNLI and QQP are excluded from our experiments.

Hyperparameters were selected via grid search, and the chosen values are detailed in Table 5. The sequence length is set to 128, with a warm-up ratio of 0.06. All tasks use a batch size of 32, trained on a single A100 40GB GPU. We fix the LoRA-XS scaling factor $\alpha$ to 16.

### C.2 INSTRUCTION TUNING EXPERIMENTS

For mathematical reasoning tasks, we perform the instruction tuning experiments on Mistral-7B-v0.1 (Jiang et al., 2023) and Gemma-7B (Team et al., 2024) decoder-only models. We use a batch size of 128 and train for 2 epochs on 100k samples of the MetaMathQA dataset. Models are evaluated on the GSM8K and MATH datasets. The learning rate is set to 4E-3 with the AdamW optimizer (Loshchilov & Hutter, 2017). The warmup ratio is 0.02, and a cosine learning rate scheduler is used. The LoRA-XS parameter $\alpha$ equals the rank.

For commonsense reasoning, the experiments are conducted on LLaMA2 7B and LLaMA3 8B decoder-only models. We train the models for 3 epochs on a collection of 170K commonsense samples (Hu et al., 2023). Models are then evaluated on BoolQ (Clark et al., 2019), PIQA (Bisk et al., 2020), SIQA (Sap et al., 2019), HellaSwag (Zellers et al., 2019), WinoGrande (Sakaguchi et al., 2021), OBQA (Mihaylov et al., 2018), ARC-c (challenge) and ARC-e (easy) datasets (Clark et al., 2018). We use a batch size of 64 and the learning rate is set to 1E-3 with the AdamW optimizer. The warmup is for 100 steps, and a linear decay is used for the learning rate scheduler. The LoRA-XS parameter $\alpha$ is set to 64.

In both mathematical and commonsense reasoning experiments, LoRA-XS modules are added to key, query, value, attention output, up projection, down projection, and gate projection layers. Two A100 80GB GPUs were used for fine-tuning.

### C.3 ABLATION EXPERIMENTAL SETUP

All ablation experiments were conducted on the GLUE benchmark using RoBERTa-large as the base model. Due to computational limits, ablations were performed on a subset of tasks. Results are reported as median values across 5 seeds, with the best model selected based on validation performance. Different learning rates were applied as detailed in the respective sections.

| Task | Rank | LoRA-XS LR | Classifier LR | Epochs |
|------|------|------------|---------------|--------|
| SST-2 | 4 | 1E-3 | 1E-3 | 20 |
| | 8 | 1E-3 | 1E-3 | 20 |
| | 12 | 5E-3 | 1E-3 | 20 |
| | 16 | 1E-3 | 5E-4 | 20 |
| | 20 | 1E-3 | 5E-3 | 20 |
| | 25 | 2E-3 | 1E-3 | 20 |
| MRPC | 4 | 1E-3 | 1E-3 | 50 |
| | 8 | 1E-3 | 6E-4 | 50 |
| | 12 | 1E-3 | 1E-3 | 50 |
| | 16 | 1E-3 | 6E-4 | 50 |
| | 20 | 1E-3 | 6E-4 | 50 |
| | 25 | 1E-3 | 6E-4 | 50 |
| CoLA | 4 | 1E-3 | 5E-3 | 50 |
| | 8 | 1E-3 | 5E-3 | 50 |
| | 12 | 1E-3 | 5E-3 | 50 |
| | 16 | 1E-3 | 1E-2 | 50 |
| | 20 | 1E-3 | 5E-3 | 50 |
| | 25 | 1E-3 | 5E-3 | 50 |
| QNLI | 4 | 1E-3 | 5E-4 | 10 |
| | 8 | 1E-3 | 1E-3 | 10 |
| | 12 | 1E-3 | 5E-4 | 10 |
| | 16 | 1E-3 | 1E-3 | 15 |
| | 20 | 1E-3 | 5E-4 | 10 |
| | 25 | 2E-3 | 6E-4 | 15 |
| RTE | 4 | 1E-3 | 6E-4 | 50 |
| | 8 | 6E-4 | 6E-4 | 50 |
| | 12 | 1E-3 | 6E-4 | 50 |
| | 16 | 1E-3 | 1E-3 | 50 |
| | 20 | 1E-3 | 1E-3 | 50 |
| | 25 | 1E-3 | 6E-4 | 50 |
| STS-B | 4 | 6E-4 | 6E-4 | 50 |
| | 8 | 1E-3 | 1E-3 | 50 |
| | 12 | 1E-3 | 1E-3 | 50 |
| | 16 | 1E-3 | 1E-3 | 50 |
| | 20 | 1E-3 | 6E-4 | 50 |
| | 25 | 1E-3 | 6E-4 | 50 |

Table 5: Hyperparameters selected via grid search for RoBERTa-large fine-tuned with LoRA-XS across various GLUE tasks. The reported hyperparameters reflect the best-performing settings found during the search.

Training was performed on a single A100 40GB GPU with a batch size of 32. We used a fixed $\alpha$ of 16, a sequence length of 128, and a warm-up ratio of 0.06. Note that MNLI initialization was not used for ablation studies.

# D ABLATION STUDY ON SINGULAR VALUE RETENTION

To gain insights into the importance of singular vectors in the pre-trained model's performance, we conduct an ablation study on fine-tuned RoBERTa-large model by performing Singular Value Decomposition (SVD) on selected weight matrices. Specifically, we decompose the matrices into their singular values and vectors, retaining only a fraction $r_{\text{frac}}$ of the singular values: either from the **top**, **middle**, or **bottom** of the spectrum, while zeroing out the remaining ones. This experiment aims to explore which portions of the singular value spectrum contribute most to the model's performance.

The experiment is carried out on RoBERTa-large fine-tuned on MRPC, SST-2, and MNLI tasks using default hyperparameters, including a learning rate of 2e-5 and 5 epochs for MRPC, 3 epochs for SST-2, and MNLI. We evaluate the performance of various weight matrices within the transformer layers, such as 'query', 'value', 'key', 'attention.output.dense', 'intermediate.dense', and 'output.dense'. Each matrix is tested across the following configurations:

- **Top-r**: Retaining the top $r_{\text{frac}}$ singular values.
- **Middle-r**: Retaining the middle $r_{\text{frac}}$ singular values.
- **Bottom-r**: Retaining the bottom $r_{\text{frac}}$ singular values.

The results, shown in Tables 6, 7, and 8, reveal several key insights. First, retaining only the top singular values consistently yields strong performance, particularly in the 'query', 'value', and 'key' matrices. Interestingly, retaining a small fraction ($r_{\text{frac}} = 0.1$) of the top singular values still achieves reasonable accuracy in tasks like SST-2 and MNLI. In contrast, retaining the middle or bottom singular values generally leads to a sharp performance degradation, suggesting their limited role in maintaining task-specific knowledge.

An exception to this pattern is observed in the 'intermediate.dense' matrix, where preserving the bottom singular values for MRPC and SST-2 yields better performance than retaining the top or middle singular values. This suggests that the 'intermediate.dense' matrix may store more task-specific information in the lower-ranked singular vectors. One possible explanation is that intermediate layers could be more sensitive to the distribution of information, requiring a broader spread across singular values, or that they store certain nuanced representations directly in the lower spectrum.

These findings provide initial evidence that the top singular vectors capture the most essential information in transformer weights. This reinforces the intuition that low-rank adaptations, such as those employed in LoRA-XS, can be highly effective in parameter-efficient fine-tuning scenarios.

| $r_{\text{frac}}$ | Module | Top $r_{\text{frac}}$ Acc | Middle $r_{\text{frac}}$ Acc | Bottom $r_{\text{frac}}$ Acc |
|---|---|---|---|---|
| 0.0 | query | 31.62 | 31.62 | 31.62 |
| 0.0 | value | 31.62 | 31.62 | 31.62 |
| 0.0 | key | 31.62 | 31.62 | 31.62 |
| 0.0 | attention.output.dense | 31.62 | 31.62 | 31.62 |
| 0.0 | intermediate.dense | 31.62 | 31.62 | 31.62 |
| 0.0 | output.dense | 31.62 | 31.62 | 31.62 |
| 0.1 | query | 70.83 | 31.62 | 31.62 |
| 0.1 | value | 69.61 | 31.62 | 31.62 |
| 0.1 | key | 55.39 | 31.62 | 31.62 |
| 0.1 | attention.output.dense | 31.86 | 31.62 | 31.62 |
| 0.1 | intermediate.dense | 31.86 | 31.62 | 68.38 |
| 0.1 | output.dense | 31.86 | 31.62 | 31.62 |
| 0.25 | query | 88.24 | 31.62 | 31.62 |
| 0.25 | value | 77.21 | 33.58 | 31.62 |
| 0.25 | key | 87.99 | 31.62 | 31.62 |
| 0.25 | attention.output.dense | 86.76 | 31.86 | 31.62 |
| 0.25 | intermediate.dense | 33.58 | 31.62 | 68.38 |
| 0.25 | output.dense | 32.11 | 31.62 | 31.62 |
| 0.5 | query | 88.24 | 31.62 | 31.62 |
| 0.5 | value | 85.05 | 31.37 | 31.62 |
| 0.5 | key | 88.97 | 31.62 | 31.62 |
| 0.5 | attention.output.dense | 88.97 | 32.60 | 31.62 |
| 0.5 | intermediate.dense | 31.62 | 31.62 | 68.38 |
| 0.5 | output.dense | 83.82 | 31.62 | 31.62 |
| 0.75 | query | 88.97 | 31.86 | 31.62 |
| 0.75 | value | 88.48 | 31.62 | 31.62 |
| 0.75 | key | 88.97 | 31.62 | 31.62 |
| 0.75 | attention.output.dense | 88.73 | 65.20 | 32.11 |
| 0.75 | intermediate.dense | 35.29 | 31.62 | 31.62 |
| 0.75 | output.dense | 86.52 | 68.38 | 31.62 |
| 0.9 | query | 88.48 | 35.78 | 32.11 |
| 0.9 | value | 88.24 | 42.89 | 31.62 |
| 0.9 | key | 88.73 | 31.86 | 31.62 |
| 0.9 | attention.output.dense | 88.73 | 84.31 | 67.65 |
| 0.9 | intermediate.dense | 37.25 | 31.62 | 68.38 |
| 0.9 | output.dense | 88.24 | 68.63 | 68.38 |
| 1.0 | query | 88.73 | 88.73 | 88.73 |
| 1.0 | value | 88.73 | 88.73 | 88.73 |
| 1.0 | key | 88.73 | 88.73 | 88.73 |
| 1.0 | attention.output.dense | 88.73 | 88.73 | 88.73 |
| 1.0 | intermediate.dense | 88.73 | 88.73 | 88.73 |
| 1.0 | output.dense | 88.73 | 88.73 | 88.73 |

Table 6: Performance on the MRPC task using varying fractions of singular values (Top, Middle, and Bottom) retained across different fine-tuned RoBERTa-large weight matrices.

| $r_{\text{frac}}$ | Module | Top $r_{\text{frac}}$ Acc | Middle $r_{\text{frac}}$ Acc | Bottom $r_{\text{frac}}$ Acc |
|---|---|---|---|---|
| 0.0 | query | 50.92 | 50.92 | 50.92 |
| 0.0 | value | 50.92 | 50.92 | 50.92 |
| 0.0 | key | 50.92 | 50.92 | 50.92 |
| 0.0 | attention.output.dense | 50.92 | 50.92 | 50.92 |
| 0.0 | intermediate.dense | 50.92 | 50.92 | 50.92 |
| 0.0 | output.dense | 49.08 | 49.08 | 49.08 |
| 0.1 | query | 93.81 | 50.92 | 50.92 |
| 0.1 | value | 91.74 | 51.26 | 49.89 |
| 0.1 | key | 93.92 | 50.92 | 50.92 |
| 0.1 | attention.output.dense | 90.60 | 55.50 | 50.92 |
| 0.1 | intermediate.dense | 50.92 | 52.18 | 50.92 |
| 0.1 | output.dense | 51.03 | 49.08 | 49.08 |
| 0.25 | query | 95.30 | 50.92 | 50.92 |
| 0.25 | value | 94.84 | 58.72 | 50.92 |
| 0.25 | key | 95.64 | 51.03 | 50.92 |
| 0.25 | attention.output.dense | 95.53 | 61.12 | 50.57 |
| 0.25 | intermediate.dense | 50.92 | 49.89 | 50.92 |
| 0.25 | output.dense | 63.19 | 53.90 | 49.08 |
| 0.5 | query | 95.64 | 51.38 | 50.92 |
| 0.5 | value | 95.76 | 53.10 | 54.70 |
| 0.5 | key | 95.64 | 51.61 | 50.92 |
| 0.5 | attention.output.dense | 95.64 | 78.78 | 63.65 |
| 0.5 | intermediate.dense | 60.55 | 49.08 | 50.92 |
| 0.5 | output.dense | 93.00 | 49.31 | 49.08 |
| 0.75 | query | 95.76 | 53.10 | 51.49 |
| 0.75 | value | 95.64 | 67.55 | 52.41 |
| 0.75 | key | 95.76 | 54.36 | 51.49 |
| 0.75 | attention.output.dense | 95.87 | 91.28 | 80.16 |
| 0.75 | intermediate.dense | 86.81 | 49.08 | 50.92 |
| 0.75 | output.dense | 95.76 | 50.92 | 49.08 |
| 0.9 | query | 95.64 | 62.04 | 55.05 |
| 0.9 | value | 95.87 | 86.93 | 73.05 |
| 0.9 | key | 95.87 | 61.58 | 55.50 |
| 0.9 | attention.output.dense | 95.87 | 94.38 | 92.89 |
| 0.9 | intermediate.dense | 89.91 | 49.08 | 49.20 |
| 0.9 | output.dense | 95.76 | 50.92 | 50.92 |
| 1.0 | query | 95.87 | 95.87 | 95.87 |
| 1.0 | value | 95.87 | 95.87 | 95.87 |
| 1.0 | key | 95.87 | 95.87 | 95.87 |
| 1.0 | attention.output.dense | 95.87 | 95.87 | 95.87 |
| 1.0 | intermediate.dense | 95.87 | 95.87 | 95.87 |
| 1.0 | output.dense | 95.87 | 95.87 | 95.87 |

Table 7: Performance on the SST2 task using varying fractions of singular values (Top, Middle, and Bottom) retained across different fine-tuned RoBERTa-large weight matrices.

| $r_{\text{frac}}$ | Module | Top $r_{\text{frac}}$ Acc | Middle $r_{\text{frac}}$ Acc | Bottom $r_{\text{frac}}$ Acc |
|---|---|---|---|---|
| 0.0 | query | 33.88 | 33.88 | 33.88 |
| 0.0 | value | 31.82 | 31.82 | 31.82 |
| 0.0 | key | 34.82 | 34.82 | 34.82 |
| 0.0 | attention.output.dense | 31.82 | 31.82 | 31.82 |
| 0.0 | intermediate.dense | 32.74 | 32.74 | 32.74 |
| 0.0 | output.dense | 31.82 | 31.82 | 31.82 |
| 0.1 | query | 68.28 | 34.19 | 34.03 |
| 0.1 | value | 81.65 | 31.82 | 31.82 |
| 0.1 | key | 70.58 | 34.49 | 34.95 |
| 0.1 | attention.output.dense | 70.19 | 31.82 | 31.82 |
| 0.1 | intermediate.dense | 33.40 | 34.29 | 33.68 |
| 0.1 | output.dense | 31.84 | 31.82 | 31.82 |
| 0.25 | query | 86.87 | 34.35 | 33.66 |
| 0.25 | value | 89.00 | 31.86 | 31.82 |
| 0.25 | key | 88.39 | 34.67 | 35.10 |
| 0.25 | attention.output.dense | 88.82 | 31.82 | 31.82 |
| 0.25 | intermediate.dense | 32.30 | 31.82 | 35.45 |
| 0.25 | output.dense | 35.62 | 31.82 | 31.82 |
| 0.5 | query | 90.07 | 34.14 | 33.83 |
| 0.5 | value | 89.79 | 31.82 | 31.80 |
| 0.5 | key | 90.22 | 35.03 | 34.98 |
| 0.5 | attention.output.dense | 90.02 | 31.78 | 31.82 |
| 0.5 | intermediate.dense | 31.85 | 32.74 | 35.45 |
| 0.5 | output.dense | 80.23 | 31.75 | 31.82 |
| 0.75 | query | 90.14 | 33.71 | 33.74 |
| 0.75 | value | 90.21 | 38.03 | 31.82 |
| 0.75 | key | 90.31 | 36.72 | 35.01 |
| 0.75 | attention.output.dense | 90.29 | 51.61 | 31.79 |
| 0.75 | intermediate.dense | 36.79 | 35.45 | 35.45 |
| 0.75 | output.dense | 89.48 | 32.33 | 32.28 |
| 0.9 | query | 90.38 | 43.21 | 34.44 |
| 0.9 | value | 90.45 | 71.56 | 41.43 |
| 0.9 | key | 90.29 | 39.03 | 37.56 |
| 0.9 | attention.output.dense | 90.30 | 83.49 | 65.65 |
| 0.9 | intermediate.dense | 38.98 | 35.45 | 35.45 |
| 0.9 | output.dense | 89.99 | 32.08 | 33.76 |
| 1.0 | query | 90.28 | 90.28 | 90.28 |
| 1.0 | value | 90.28 | 90.28 | 90.28 |
| 1.0 | key | 90.28 | 90.28 | 90.28 |
| 1.0 | attention.output.dense | 90.28 | 90.28 | 90.28 |
| 1.0 | intermediate.dense | 90.28 | 90.28 | 90.28 |
| 1.0 | output.dense | 90.28 | 90.28 | 90.28 |

Table 8: Performance on the MNLI task using varying fractions of singular values (Top, Middle, and Bottom) retained across different fine-tuned RoBERTa-large weight matrices.

# E   ABLATION STUDY: EFFECT OF RETAINING SINGULAR VECTOR SUBSPACES ON DELTA WEIGHT APPROXIMATION

In this ablation study, we explore the influence of retaining different singular vector subspaces on the approximation of weight updates $\Delta W$ in LLM fine-tuning. Specifically, we assess how well $\Delta W$, representing the full weight updates, can be approximated by projecting it onto a subspace spanned by a subset of the singular vectors obtained from the SVD of the pre-trained model's weights $W$. Our goal is to evaluate how different subspaces (top, middle, or bottom) affect downstream performance across several GLUE tasks.

## E.1   THEORETICAL FRAMEWORK

The pre-trained model's weights $W$ are decomposed using singular value decomposition (SVD) as follows:

$$W = U\Sigma V^T$$

where $U$ and $V$ are orthogonal matrices representing the left and right singular vectors, respectively, and $\Sigma$ is the diagonal matrix of singular values. During fine-tuning or LoRA adaptation, the model's weights are updated, resulting in a weight update $\Delta W$. This leads to the updated weight matrix being expressed as:

$$W + \Delta W = U(\Sigma + C)V^T$$

where $\Delta W$ can be represented in the singular vector basis of $W$ as:

$$\Delta W = UCV^T, \quad C = U^T \Delta W V$$

The matrix $C$ captures how the weight update $\Delta W$ projects onto the singular vector subspaces of $W$. We propose approximating $\Delta W$ by retaining only specific subspaces of $U$ and $V$, focusing on subspaces corresponding to the top, middle, or bottom singular vectors. This leads to the following approximation:

$$\Delta W_{\text{approx}} = UC_r V^T$$

where $C_r$ is an $r \times r$ block matrix, constructed by selecting singular vector subspaces (top, middle, or bottom). By retaining different subspaces, we explore the sensitivity of the model's performance to various parts of the singular spectrum.

## E.2   FULLY FINE-TUNED $\Delta W$

In this experiment, $\Delta W$ represents the complete set of weight updates from fine-tuning. We fine-tuned RoBERTa-large on MRPC, SST-2, and MNLI, using a learning rate of 2e-5 for 5 epochs on MRPC and 3 epochs on SST-2 and MNLI. This experiment aims to assess the impact of retaining different fractions of the singular spectrum on performance for various weight modules, including query, key, value, intermediate dense, and output dense. The results are summarized in Table 9.

Our findings reveal that different modules exhibit varying sensitivity to the retention of singular vectors, suggesting that task-specific fine-tuning impacts each module differently. In the self-attention modules (query, key, value, and attention output), retaining even a small fraction of singular vectors (1% or 10%) resulted in minimal performance degradation across all tasks (in Table 9 we only show results for the query because for other matrices the behavior was very similar). Both the top and bottom singular vectors maintained performance well. This may indicate that $\Delta W$ is relatively small for self-attention layers compared to $W$, meaning it has a limited influence on the model's predictions. This may suggest that these modules can tolerate substantial dimensionality reduction. The intermediate dense modules exhibited slightly higher sensitivity to the subspaces retained, though the impact remained small.

The output dense modules demonstrated the greatest sensitivity to SVD-based approximations. This increased sensitivity in the output dense layers may indicate that $\Delta W/W$ may be larger in these layers. Consequently, we hypothesize that higher ranks should be used for the output dense layers, while lower ranks could be sufficient for the self-attention layers.

| Module | $r_{\text{frac}}$ | retain | MRPC | SST-2 | MNLI |
|--------|------|--------|------|-------|------|
| query | 0.01 | top | 89.71 | 95.64 | 90.08 |
| | 0.01 | middle | 89.71 | 95.64 | 90.05 |
| | 0.01 | bottom | 89.71 | 95.64 | 90.05 |
| | 0.1 | top | 89.71 | 95.64 | 90.07 |
| | 0.1 | middle | 89.71 | 95.64 | 90.05 |
| | 0.1 | bottom | 89.71 | 95.53 | 90.10 |
| | 0.25 | top | 89.71 | 95.64 | 90.14 |
| | 0.25 | middle | 89.71 | 95.53 | 90.08 |
| | 0.25 | bottom | 89.71 | 95.53 | 90.14 |
| | 0.5 | top | 89.46 | 95.64 | 90.20 |
| | 0.5 | middle | 89.71 | 95.53 | 90.16 |
| | 0.5 | bottom | 89.71 | 95.76 | 90.16 |
| intermediate.dense | 0.01 | top | 88.97 | 95.53 | 88.16 |
| | 0.01 | middle | 88.48 | 95.53 | 88.45 |
| | 0.01 | bottom | 88.73 | 95.53 | 88.46 |
| | 0.1 | top | 88.73 | 95.41 | 88.16 |
| | 0.1 | middle | 88.48 | 95.53 | 88.46 |
| | 0.1 | bottom | 88.73 | 95.53 | 88.46 |
| | 0.25 | top | 89.22 | 95.41 | 88.18 |
| | 0.25 | middle | 88.73 | 95.53 | 88.48 |
| | 0.25 | bottom | 88.48 | 95.53 | 88.55 |
| | 0.5 | top | 89.22 | 95.41 | 88.40 |
| | 0.5 | middle | 88.48 | 95.53 | 88.60 |
| | 0.5 | bottom | 88.73 | 95.53 | 88.58 |
| output.dense | 0.01 | top | 79.41 | 94.04 | 86.82 |
| | 0.01 | middle | 79.41 | 94.04 | 86.84 |
| | 0.01 | bottom | 79.41 | 94.04 | 86.84 |
| | 0.1 | top | 80.15 | 94.61 | 87.05 |
| | 0.1 | middle | 79.41 | 94.04 | 86.82 |
| | 0.1 | bottom | 80.64 | 94.72 | 87.17 |
| | 0.25 | top | 81.37 | 95.18 | 88.76 |
| | 0.25 | middle | 79.90 | 94.27 | 87.05 |
| | 0.25 | bottom | 84.31 | 95.53 | 88.26 |
| | 0.5 | top | 86.27 | 95.53 | 89.34 |
| | 0.5 | middle | 80.88 | 94.61 | 87.72 |
| | 0.5 | bottom | 87.75 | 95.87 | 89.01 |

Table 9: Performance results of the SVD-based delta weight approximation experiment across different tasks (MRPC, SST-2, MNLI) and modules. The table reports the performance for various fractions of retained singular vectors ($r_{\text{frac}}$) from the top, middle, and bottom subspaces.

# F  ABLATION STUDY ON LoRA-XS INITIALIZATION

In this ablation study, we examine the impact of different initialization strategies for the LoRA-XS matrices $A$ and $B$, comparing Singular Value Decomposition (SVD) initialization to random initialization (used in LoRA and VeRA). Additionally, we investigate whether applying SVD to random matrices yields any advantages, testing if the benefits of SVD initialization derive solely from the orthogonality of singular vectors, regardless of the source of the decomposition.

For the SVD initialization, we decompose the original weight matrix as $W \approx U_r \Sigma_r V_r^T$, retaining the top $r$ singular vectors. Random initialization follows the Kaiming initialization, commonly used for linear layers in PyTorch (Paszke et al., 2019).

We conduct our experiments using the RoBERTa-large (Liu et al., 2019) model on selected GLUE tasks. The LoRA-XS matrices are applied to specific transformer weights matrices, including the query, value, attention output, and intermediate output layers. All experiments are repeated with five different random seeds, and we report the median performance.

## F.1  COMPARING RANDOM INITIALIZATION, SVD ON RANDOM MATRICES, AND SVD ON WEIGHTS

We compare three initialization strategies: random initialization, SVD on random matrices (SVD of random), and SVD on the corresponding layer's weight matrix (SVD of W). The results, summarized in Table 11, Table 12, and Table 13, reveal that initializing the $A$ and $B$ matrices with SVD on the corresponding module's original weight matrix ($W$) generally provides the best performance. This supports our theoretical hypothesis that leveraging the structure of the weight matrix through SVD initialization is more effective than random initialization, as it retains task-relevant information from the pretrained model (see Section 3.1).

Our results also confirm that the performance benefits do not stem solely from the orthogonality of the singular vectors, as SVD on random matrices did not consistently outperform random initialization.

Notably, for the SST-2 task (Table 10), all initialization methods perform similarly, with SVD on random matrices yielding the highest score. This is likely due to the nature of SST-2 as a sentiment classification task, which is less closely aligned with language modeling compared to other tasks such as MRPC, COLA, and QNLI. This finding aligns with our theoretical analysis (Section 3.1), which suggests that SVD on the original weights is most beneficial when the fine-tuning task is similar to the pretraining objective.

## F.2  EFFECT OF INITIALIZATION ON LEARNING CURVE AND RANK

We further explore the impact of initialization on learning speed and performance at different ranks.

In Table 14, we present the detailed results comparing the initialization of the $A$ and $B$ matrices of LoRA-XS using SVD versus random initialization. The reported metrics are accuracy for SST-2 and QNLI, and Matthews correlation for CoLA on the RoBERTa-large model. Hyperparameters were selected through grid search and are detailed in Table 15. Results are reported as the median over 5 random seeds, with the best epoch result for each run (column "Best epoch"). Additionally, we show results after 1 and 2 epochs to illustrate the learning curve and how quickly the models learn with each initialization method.

Our results empirically confirm that also for other ranks aligning adaptation matrices with the principal components of pre-trained weights can enhance parameter efficiency while maintaining model accuracy. As shown in Table 14, initializing the $A$ and $B$ matrices with SVD generally leads to improved performance compared to random initialization.

Moreover, the performance after 1 and 2 epochs is often higher with SVD initialization. This suggests that proper initialization using SVD may help the models converge more quickly and achieve better overall accuracy. This finding underscores the importance of initialization strategy in fine-tuning large language models, highlighting that SVD initialization not only boosts performance but also accelerates the training process.

| Rank | Init Type | LR | CLS LR | Median Score | Std Dev |
|------|-----------|-----|--------|--------------|---------|
| 4 | random | 0.0005 | 0.0005 | 94.38 | 0.37 |
| 4 | random | 0.0005 | 0.001 | 94.50 | 0.37 |
| 4 | random | 0.0005 | 0.005 | 94.61 | 0.45 |
| 4 | random | 0.001 | 0.0005 | 94.61 | 0.41 |
| 4 | random | 0.001 | 0.001 | 94.72 | 0.36 |
| 4 | random | 0.001 | 0.005 | 94.61 | 0.33 |
| 4 | random | 0.005 | 0.0005 | 94.38 | 0.34 |
| 4 | random | 0.005 | 0.001 | 94.61 | 0.53 |
| 4 | random | 0.005 | 0.005 | 94.15 | 0.44 |
| 4 | SVD of random | 0.0005 | 0.0005 | 94.15 | 0.81 |
| 4 | SVD of random | 0.0005 | 0.001 | 94.04 | 0.63 |
| 4 | SVD of random | 0.0005 | 0.005 | 93.92 | 0.92 |
| 4 | SVD of random | 0.001 | 0.0005 | 94.38 | 0.65 |
| 4 | SVD of random | 0.001 | 0.001 | 94.27 | 0.66 |
| 4 | SVD of random | 0.001 | 0.005 | 94.04 | 0.85 |
| 4 | SVD of random | 0.005 | 0.0005 | 94.72 | 0.54 |
| 4 | **SVD of random** | 0.005 | 0.001 | **94.84** | 0.48 |
| 4 | SVD of random | 0.005 | 0.005 | 94.50 | 0.50 |
| 4 | SVD of W | 0.0005 | 0.0005 | 94.50 | 0.22 |
| 4 | SVD of W | 0.0005 | 0.001 | 94.15 | 0.41 |
| 4 | SVD of W | 0.0005 | 0.005 | 94.38 | 0.21 |
| 4 | SVD of W | 0.001 | 0.0005 | 94.72 | 0.19 |
| 4 | SVD of W | 0.001 | 0.001 | 94.38 | 0.09 |
| 4 | SVD of W | 0.001 | 0.005 | 94.50 | 0.39 |
| 4 | SVD of W | 0.005 | 0.0005 | 50.92 | 20.09 |
| 4 | SVD of W | 0.005 | 0.001 | 88.99 | 19.32 |
| 4 | SVD of W | 0.005 | 0.005 | 91.40 | 15.97 |

Table 10: Median accuracy scores for SST-2 across different initialization methods (rank 4). Despite similar performance across initialization strategies, the best score is achieved using SVD applied to random matrices, likely due to the task's nature as a sentiment classification challenge.

| Rank | Init Type | LR | CLS LR | Median Score | Std Dev |
|---|---|---|---|---|---|
| 4 | random | 0.0005 | 0.0005 | 82.60 | 0.63 |
| 4 | random | 0.0005 | 0.001 | 82.84 | 0.93 |
| 4 | random | 0.0005 | 0.005 | 83.33 | 0.78 |
| 4 | random | 0.001 | 0.0005 | 84.80 | 0.63 |
| 4 | random | 0.001 | 0.001 | 84.80 | 1.04 |
| 4 | random | 0.001 | 0.005 | 84.80 | 1.25 |
| 4 | random | 0.005 | 0.0005 | 86.76 | 0.70 |
| 4 | random | 0.005 | 0.001 | 85.78 | 0.80 |
| 4 | random | 0.005 | 0.005 | 86.76 | 0.96 |
| 4 | SVD of random | 0.0005 | 0.0005 | 78.92 | 1.25 |
| 4 | SVD of random | 0.0005 | 0.001 | 79.41 | 1.36 |
| 4 | SVD of random | 0.0005 | 0.005 | 78.43 | 1.05 |
| 4 | SVD of random | 0.001 | 0.0005 | 81.62 | 0.77 |
| 4 | SVD of random | 0.001 | 0.001 | 81.13 | 1.19 |
| 4 | SVD of random | 0.001 | 0.005 | 80.64 | 1.61 |
| 4 | SVD of random | 0.005 | 0.0005 | 84.31 | 0.74 |
| 4 | SVD of random | 0.005 | 0.001 | 84.31 | 1.07 |
| 4 | SVD of random | 0.005 | 0.005 | 84.56 | 1.27 |
| 4 | SVD of W | 0.0005 | 0.0005 | 86.76 | 0.68 |
| 4 | SVD of W | 0.0005 | 0.001 | 86.52 | 0.33 |
| 4 | SVD of W | 0.0005 | 0.005 | 86.03 | 0.87 |
| 4 | SVD of W | 0.001 | 0.0005 | 87.50 | 0.73 |
| 4 | SVD of W | 0.001 | 0.001 | 87.25 | 0.72 |
| 4 | **SVD of W** | 0.001 | 0.005 | **87.50** | 0.99 |
| 4 | SVD of W | 0.005 | 0.0005 | 69.12 | 2.13 |
| 4 | SVD of W | 0.005 | 0.001 | 70.59 | 2.05 |
| 4 | SVD of W | 0.005 | 0.005 | 68.38 | 2.27 |

Table 11: Median accuracy scores for MRPC across different initialization methods (rank 4). SVD applied to the original weight matrix provides the best performance, aligning with the hypothesis that initializing based on weight-specific information better adapts LoRA-XS to the task.

| Rank | Init Type | LR | CLS LR | Median Score | Std Dev |
|---|---|---|---|---|---|
| 4 | random | 0.0005 | 0.0005 | 51.88 | 0.83 |
| 4 | random | 0.0005 | 0.001 | 53.12 | 1.05 |
| 4 | random | 0.0005 | 0.005 | 54.81 | 2.08 |
| 4 | random | 0.001 | 0.0005 | 52.90 | 1.63 |
| 4 | random | 0.001 | 0.001 | 53.38 | 1.12 |
| 4 | random | 0.001 | 0.005 | 58.53 | 1.84 |
| 4 | random | 0.005 | 0.0005 | 52.48 | 1.44 |
| 4 | random | 0.005 | 0.001 | 53.17 | 1.88 |
| 4 | random | 0.005 | 0.005 | 56.34 | 1.15 |
| 4 | SVD of random | 0.0005 | 0.0005 | 47.96 | 1.52 |
| 4 | SVD of random | 0.0005 | 0.001 | 48.58 | 2.02 |
| 4 | SVD of random | 0.0005 | 0.005 | 53.30 | 0.92 |
| 4 | SVD of random | 0.001 | 0.0005 | 50.14 | 1.88 |
| 4 | SVD of random | 0.001 | 0.001 | 51.47 | 2.06 |
| 4 | SVD of random | 0.001 | 0.005 | 55.27 | 0.96 |
| 4 | SVD of random | 0.005 | 0.0005 | 52.33 | 2.72 |
| 4 | SVD of random | 0.005 | 0.001 | 53.32 | 2.16 |
| 4 | SVD of random | 0.005 | 0.005 | 56.33 | 1.21 |
| 4 | SVD of W | 0.0005 | 0.0005 | 55.99 | 0.55 |
| 4 | SVD of W | 0.0005 | 0.001 | 55.74 | 1.17 |
| 4 | SVD of W | 0.0005 | 0.005 | 58.20 | 1.66 |
| 4 | SVD of W | 0.001 | 0.0005 | 57.63 | 0.78 |
| 4 | SVD of W | 0.001 | 0.001 | 56.76 | 0.66 |
| 4 | **SVD of W** | 0.001 | 0.005 | **60.11** | 0.85 |
| 4 | SVD of W | 0.005 | 0.0005 | 43.10 | 7.78 |
| 4 | SVD of W | 0.005 | 0.001 | 38.02 | 16.30 |
| 4 | SVD of W | 0.005 | 0.005 | 45.87 | 8.41 |

Table 12: Median performance scores (Matthews correlation) for COLA across different initialization methods (rank 4). SVD applied to the original weight matrix provides the best performance, aligning with the hypothesis that initializing based on weight-specific information better adapts LoRA-XS to the task.

| Rank | Init Type | LR | CLS LR | Median Score | Std Dev |
|------|-----------|--------|--------|--------------|---------|
| 4 | random | 0.0005 | 0.0005 | 87.92 | 0.56 |
| 4 | random | 0.0005 | 0.001 | 87.64 | 0.31 |
| 4 | random | 0.0005 | 0.005 | 87.79 | 0.32 |
| 4 | random | 0.001 | 0.0005 | 88.17 | 0.44 |
| 4 | random | 0.001 | 0.001 | 88.45 | 0.27 |
| 4 | random | 0.001 | 0.005 | 87.96 | 0.46 |
| 4 | random | 0.005 | 0.0005 | 88.91 | 0.56 |
| 4 | random | 0.005 | 0.001 | 88.80 | 0.42 |
| 4 | random | 0.005 | 0.005 | 88.32 | 0.35 |
| 4 | SVD of random | 0.0005 | 0.0005 | 84.48 | 1.14 |
| 4 | SVD of random | 0.0005 | 0.001 | 84.42 | 1.00 |
| 4 | SVD of random | 0.0005 | 0.005 | 83.12 | 0.98 |
| 4 | SVD of random | 0.001 | 0.0005 | 86.36 | 0.88 |
| 4 | SVD of random | 0.001 | 0.001 | 86.18 | 1.01 |
| 4 | SVD of random | 0.001 | 0.005 | 85.30 | 0.97 |
| 4 | SVD of random | 0.005 | 0.0005 | 88.36 | 0.58 |
| 4 | SVD of random | 0.005 | 0.001 | 88.34 | 0.42 |
| 4 | SVD of random | 0.005 | 0.005 | 87.33 | 0.64 |
| 4 | SVD of W | 0.0005 | 0.0005 | 90.74 | 0.19 |
| 4 | SVD of W | 0.0005 | 0.001 | 90.74 | 0.13 |
| 4 | SVD of W | 0.0005 | 0.005 | 90.41 | 0.10 |
| 4 | **SVD of W** | 0.001 | 0.0005 | **90.94** | 0.20 |
| 4 | SVD of W | 0.001 | 0.001 | 90.92 | 0.17 |
| 4 | SVD of W | 0.001 | 0.005 | 90.59 | 0.90 |
| 4 | SVD of W | 0.005 | 0.0005 | 50.54 | 14.37 |
| 4 | SVD of W | 0.005 | 0.001 | 50.54 | 0.00 |
| 4 | SVD of W | 0.005 | 0.005 | 50.54 | 0.00 |

Table 13: Median accuracy scores for QNLI across different initialization methods (rank 4). SVD applied to the original weight matrix provides the best performance, aligning with the hypothesis that initializing based on weight-specific information better adapts LoRA-XS to the task.

| Task | Rank | Init Type | Performance | | |
|------|------|-----------|-------------|--|--|
| | | | After 1 epoch | After 2 epoch | Best epoch |
| Cola | 8 | Random | $0 \pm 17.59$ | $40.91 \pm 15.57$ | $60.29 \pm 1.22$ |
| | | SVD of W | $\mathbf{12.59} \pm 11.09$ | $\mathbf{43.84} \pm 15.62$ | $\mathbf{64.39} \pm 0.75$ |
| | 12 | Random | $5.8 \pm 14.71$ | $26.77 \pm 20.43$ | $62.96 \pm 1.25$ |
| | | SVD of W | $\mathbf{13.16} \pm 17.84$ | $\mathbf{44.22} \pm 13.04$ | $\mathbf{65.47} \pm 0.90$ |
| | 20 | Random | $\mathbf{47.17} \pm 2.59$ | $43.42 \pm 5.70$ | $64.7 \pm 1.10$ |
| | | SVD of W | $18.83 \pm 9.47$ | $\mathbf{44.77} \pm 9.65$ | $\mathbf{68.08} \pm 1.21$ |
| SST-2 | 8 | Random | $92.55 \pm 0.74$ | $93.58 \pm 0.53$ | $\mathbf{95.30} \pm 0.34$ |
| | | SVD of W | $\mathbf{93.23} \pm 0.27$ | $\mathbf{93.81} \pm 0.22$ | $94.95 \pm 0.07$ |
| | 12 | Random | $\mathbf{93.46} \pm 0.54$ | $\mathbf{94.27} \pm 0.46$ | $95.64 \pm 0.54$ |
| | | SVD of W | $\mathbf{93.46} \pm 0.86$ | $93.35 \pm 0.49$ | $\mathbf{95.87} \pm 0.28$ |
| | 20 | Random | $94.15 \pm 0.67$ | $\mathbf{94.15} \pm 0.58$ | $\mathbf{95.87} \pm 0.24$ |
| | | SVD of W | $\mathbf{94.61} \pm 0.46$ | $93.92 \pm 0.87$ | $\mathbf{95.87} \pm 0.31$ |
| QNLI | 8 | Random | $88.5 \pm 0.86$ | $89.99 \pm 0.72$ | $91.4 \pm 0.59$ |
| | | SVD of W | $\mathbf{89.84} \pm 0.37$ | $\mathbf{91.31} \pm 0.10$ | $\mathbf{92.49} \pm 0.09$ |
| | 12 | Random | $89.27 \pm 0.55$ | $90.3 \pm 0.58$ | $92.51 \pm 0.42$ |
| | | SVD of W | $\mathbf{90.74} \pm 0.25$ | $\mathbf{92.17} \pm 0.53$ | $\mathbf{93.32} \pm 0.51$ |
| | 20 | Random | $89.73 \pm 0.96$ | $91.45 \pm 0.11$ | $93.23 \pm 0.30$ |
| | | SVD of W | $\mathbf{91.49} \pm 0.18$ | $\mathbf{92.24} \pm 0.24$ | $\mathbf{94.05} \pm 0.16$ |

Table 14: The impact of random vs SVD of W initialization scheme on the performance of RoBERTa-large on three GLUE tasks across different ranks. Matrices $A$ and $B$ in LoRA-XS are initialized either randomly or using SVD of the corresponding weight. We report Matthew's Correlation for CoLA and accuracy for SST-2 and QNLI tasks. Each table entry reports median and standard deviation of five runs with different seeds. Our study indicates that the SVD-based initialization generally results in better performance, especially when dealing with lower ranks (*i.e.*, less trainable parameters).

| Task | Init | Rank | LoRA-XS LR | Classifier LR | Epochs |
|------|------|------|------------|---------------|--------|
| SST-2 | SVD of W | 8 | 1E-3 | 1E-3 | 20 |
| | | 12 | 5E-3 | 1E-3 | 20 |
| | | 20 | 1E-3 | 5E-3 | 20 |
| | random | 8 | 1E-3 | 1E-3 | 20 |
| | | 12 | 1E-3 | 1E-3 | 20 |
| | | 20 | 5E-3 | 1E-4 | 20 |
| CoLA | SVD of W | 8 | 1E-3 | 5E-3 | 50 |
| | | 12 | 1E-3 | 5E-3 | 50 |
| | | 20 | 1E-3 | 5E-3 | 50 |
| | random | 8 | 1E-3 | 1E-2 | 50 |
| | | 12 | 1E-3 | 1E-2 | 50 |
| | | 20 | 5E-3 | 5E-3 | 50 |
| QNLI | SVD of W | 8 | 1E-3 | 1E-3 | 10 |
| | | 12 | 1E-3 | 5E-4 | 10 |
| | | 20 | 1E-3 | 5E-4 | 10 |
| | random | 8 | 1E-3 | 1E-3 | 10 |
| | | 12 | 1E-3 | 5E-4 | 10 |
| | | 20 | 1E-3 | 1E-3 | 10 |

Table 15: Hyperparameters for the ablation study with different $A$ and $B$ initialization methods (SVD and random) on RoBERTa-large with LoRA-XS across various GLUE tasks.

# G    LoRA-XS USING TOP VERSUS BOTTOM SINGULAR VECTORS

In this ablation study, we evaluate the effect of initializing LoRA-XS with either top or bottom singular vectors derived from the singular value decomposition (SVD) of the weight matrix $W$. We conduct experiments on the RoBERTa-large model across multiple GLUE benchmark tasks, including CoLA, QNLI, MRPC, and SST-2. The target layers for LoRA-XS in these experiments are the query, value, attention.output.dense, and output.dense layers. We set the LoRA scaling factor $\alpha$ to 16 and the rank $r$ to 4.

Given a weight matrix $W \in \mathbb{R}^{m \times n}$, we compute its SVD as $W = U\Sigma V^T$, where $U \in \mathbb{R}^{m \times m}$, $\Sigma \in \mathbb{R}^{m \times n}$ is a diagonal matrix of singular values, and $V \in \mathbb{R}^{n \times n}$.

For LoRA-XS initialization, we modify the decomposition by selecting either the top or bottom $r$ singular vectors from $U$ and $V$:

- **Top SVD initialization:** We select the first $r$ singular vectors from $U$ and $V$. Specifically, $A = U_r\Sigma_r$ and $B = V_r^T$, where $U_r \in \mathbb{R}^{m \times r}$ and $V_r \in \mathbb{R}^{n \times r}$ represent the top $r$ singular vectors.

- **Bottom SVD initialization:** We select the last $r$ singular vectors from $U$ and $V$, i.e., $A = U_{-r}\Sigma_{-r}$ and $B = V_{-r}^T$.

Each configuration is tested over five random seeds, and we report the median performance. The results are summarized in Table 16, Table 17, Table 18, and Table 19.

Across all tasks, initializing LoRA-XS with top singular vectors consistently outperforms initialization with bottom singular vectors. These results support our decision to adopt top singular vector initialization for LoRA-XS.

| Rank | Init Type | LR | CLS LR | Median Score | Std Dev |
|------|-----------|-----|--------|--------------|---------|
| 4 | svd on w bottom | 0.0005 | 0.0005 | 42.26 | 1.71 |
| 4 | svd on w bottom | 0.0005 | 0.001 | 43.28 | 1.46 |
| 4 | svd on w bottom | 0.0005 | 0.005 | 46.09 | 1.26 |
| 4 | svd on w bottom | 0.001 | 0.0005 | 45.24 | 1.71 |
| 4 | svd on w bottom | 0.001 | 0.001 | 44.53 | 0.60 |
| 4 | svd on w bottom | 0.001 | 0.005 | 48.24 | 0.62 |
| 4 | svd on w bottom | 0.005 | 0.0005 | 47.04 | 1.06 |
| 4 | svd on w bottom | 0.005 | 0.001 | 48.89 | 0.83 |
| 4 | svd on w bottom | 0.005 | 0.005 | 52.01 | 0.92 |
| 4 | svd on w top | 0.0005 | 0.0005 | 55.01 | 0.58 |
| 4 | svd on w top | 0.0005 | 0.001 | 55.47 | 0.63 |
| 4 | svd on w top | 0.0005 | 0.005 | 58.63 | 1.21 |
| 4 | svd on w top | 0.001 | 0.0005 | 58.04 | 1.31 |
| 4 | svd on w top | 0.001 | 0.001 | 57.46 | 0.75 |
| 4 | **svd on w top** | 0.001 | 0.005 | **60.29** | 1.06 |
| 4 | svd on w top | 0.005 | 0.0005 | 43.78 | 14.36 |
| 4 | svd on w top | 0.005 | 0.001 | 47.60 | 3.72 |
| 4 | svd on w top | 0.005 | 0.005 | 25.51 | 15.58 |

Table 16: Results for CoLA task comparing LoRA-XS initialized with top versus bottom singular vectors across the query, value, attention.output.dense, and "output.dense" modules in RoBERTa-large model. Top singular vectors demonstrate superior performance.

| Rank | Init Type | LR | CLS LR | Median Score | Std Dev |
|---|---|---|---|---|---|
| 4 | svd on w bottom | 0.0005 | 0.0005 | 73.88 | 0.27 |
| 4 | svd on w bottom | 0.0005 | 0.001 | 73.24 | 0.23 |
| 4 | svd on w bottom | 0.0005 | 0.005 | 66.17 | 1.67 |
| 4 | svd on w bottom | 0.001 | 0.0005 | 77.52 | 0.25 |
| 4 | svd on w bottom | 0.001 | 0.001 | 77.14 | 0.33 |
| 4 | svd on w bottom | 0.001 | 0.005 | 70.14 | 1.13 |
| 4 | svd on w bottom | 0.005 | 0.0005 | 81.29 | 0.29 |
| 4 | svd on w bottom | 0.005 | 0.001 | 81.38 | 0.22 |
| 4 | svd on w bottom | 0.005 | 0.005 | 79.90 | 0.33 |
| 4 | svd on w top | 0.0005 | 0.0005 | 90.88 | 0.15 |
| 4 | svd on w top | 0.0005 | 0.001 | 90.68 | 0.05 |
| 4 | svd on w top | 0.0005 | 0.005 | 90.28 | 0.20 |
| 4 | **svd on w top** | 0.001 | 0.0005 | **90.98** | 0.09 |
| 4 | svd on w top | 0.001 | 0.001 | 90.96 | 0.11 |
| 4 | svd on w top | 0.001 | 0.005 | 90.72 | 0.21 |
| 4 | svd on w top | 0.005 | 0.0005 | 50.54 | 0.75 |
| 4 | svd on w top | 0.005 | 0.001 | 50.54 | 0.00 |
| 4 | svd on w top | 0.005 | 0.005 | 50.54 | 0.00 |

Table 17: Results for QNLI task comparing LoRA-XS initialized with top versus bottom singular vectors across the query, value, attention.output.dense, and "output.dense" modules in RoBERTa-large model. Top singular vectors demonstrate superior performance.

| Rank | Init Type | LR | CLS LR | Median Score | Std Dev |
|---|---|---|---|---|---|
| 4 | svd on w bottom | 0.0005 | 0.0005 | 75.25 | 0.66 |
| 4 | svd on w bottom | 0.0005 | 0.001 | 75.49 | 0.42 |
| 4 | svd on w bottom | 0.0005 | 0.005 | 74.51 | 0.48 |
| 4 | svd on w bottom | 0.001 | 0.0005 | 75.00 | 0.50 |
| 4 | svd on w bottom | 0.001 | 0.001 | 75.49 | 0.47 |
| 4 | svd on w bottom | 0.001 | 0.005 | 75.00 | 0.29 |
| 4 | svd on w bottom | 0.005 | 0.0005 | 78.92 | 0.48 |
| 4 | svd on w bottom | 0.005 | 0.001 | 79.17 | 0.48 |
| 4 | svd on w bottom | 0.005 | 0.005 | 77.94 | 0.41 |
| 4 | svd on w top | 0.0005 | 0.0005 | 86.27 | 0.63 |
| 4 | svd on w top | 0.0005 | 0.001 | 85.54 | 1.01 |
| 4 | svd on w top | 0.0005 | 0.005 | 85.54 | 1.23 |
| 4 | svd on w top | 0.001 | 0.0005 | 86.52 | 0.73 |
| 4 | **svd on w top** | 0.001 | 0.001 | **86.76** | 1.11 |
| 4 | **svd on w top** | 0.001 | 0.005 | **86.76** | 0.81 |
| 4 | svd on w top | 0.005 | 0.0005 | 69.85 | 6.25 |
| 4 | svd on w top | 0.005 | 0.001 | 74.26 | 6.69 |
| 4 | svd on w top | 0.005 | 0.005 | 68.63 | 1.79 |

Table 18: Results for MRPC task comparing LoRA-XS initialized with top versus bottom singular vectors across the query, value, attention.output.dense, and "output.dense" modules in RoBERTa-large model. Top singular vectors demonstrate superior performance.

| Rank | Init Type | LR | CLS LR | Median Score | Std Dev |
|------|-----------|----|--------|--------------|---------|
| 4 | svd on w bottom | 0.0005 | 0.0005 | 89.11 | 0.16 |
| 4 | svd on w bottom | 0.0005 | 0.001 | 89.45 | 0.28 |
| 4 | svd on w bottom | 0.0005 | 0.005 | 88.76 | 0.09 |
| 4 | svd on w bottom | 0.001 | 0.0005 | 90.48 | 0.11 |
| 4 | svd on w bottom | 0.001 | 0.001 | 90.94 | 0.22 |
| 4 | svd on w bottom | 0.001 | 0.005 | 90.08 | 2.04 |
| 4 | svd on w bottom | 0.005 | 0.0005 | 92.89 | 0.28 |
| 4 | svd on w bottom | 0.005 | 0.001 | 93.12 | 0.12 |
| 4 | svd on w bottom | 0.005 | 0.005 | 92.66 | 0.12 |
| 4 | **svd on w top** | 0.0005 | 0.0005 | **94.61** | 0.17 |
| 4 | svd on w top | 0.0005 | 0.001 | 94.38 | 0.15 |
| 4 | svd on w top | 0.0005 | 0.005 | 94.50 | 0.15 |
| 4 | svd on w top | 0.001 | 0.0005 | 94.50 | 0.14 |
| 4 | svd on w top | 0.001 | 0.001 | 94.38 | 0.16 |
| 4 | svd on w top | 0.001 | 0.005 | 94.21 | 0.10 |
| 4 | svd on w top | 0.005 | 0.0005 | 91.86 | 20.21 |
| 4 | svd on w top | 0.005 | 0.001 | 91.28 | 16.27 |
| 4 | svd on w top | 0.005 | 0.005 | 50.92 | 19.47 |

Table 19: Results for SST-2 task comparing LoRA-XS initialized with top versus bottom singular vectors across the query, value, attention.output.dense, and "output.dense" modules in RoBERTa-large model. Top singular vectors demonstrate superior performance.

# H  THE IMPORTANCE OF SINGULAR VALUES FOR LORA-XS

In this section, we explore the significance of including singular values in the initialization of matrix $A$ in LoRA-XS. Specifically, we compare two initialization methods: one where matrix $A$ is initialized with both singular vectors and singular values, $A = U\Sigma$, and another where matrix $A$ is initialized using only the singular vectors, $A = U$, while keeping $B = V^T$ in both cases. Our analysis focuses on the top $r$ singular vectors for both methods.

For LoRA-XS, we initialize the matrices $A$ and $B$ as:

$$A = U_r\Sigma_r \quad \text{and} \quad B = V_r^T \tag{7}$$

In the variant without singular values, matrix $A$ is initialized as:

$$A = U_r \quad \text{and} \quad B = V_r^T \tag{8}$$

We conduct experiments using the RoBERTa-large model across several GLUE benchmark tasks, including CoLA, QNLI, MRPC, and SST-2. The target layers for LoRA-XS are the query, value, attention.output.dense, and output.dense layers. We set the LoRA scaling factor $\alpha$ to 16 and use a rank of 4. Each configuration is tested over 5 random seeds, and we report the median performance. The results are presented in Table 20, Table 21, Table 22, and Table 23.

Overall, our results indicate that including singular values ($A = U_r\Sigma_r$) provides consistent benefits across most tasks, particularly for CoLA, QNLI, and SST-2. However, for the MRPC task, initializing matrix $A$ without the singular values ($A = U_r$) achieves better performance. The findings support the inclusion of singular values in most cases, but also demonstrate that certain tasks, like MRPC, may benefit from simpler initialization methods.

| Rank | Init Type | LR | CLS LR | Median Score | Std Dev |
|---|---|---|---|---|---|
| 4 | $A = U_r\Sigma_r$ | 0.0005 | 0.0005 | **94.61** | 0.17 |
| 4 | $A = U_r\Sigma_r$ | 0.0005 | 0.001 | 94.38 | 0.15 |
| 4 | $A = U_r\Sigma_r$ | 0.0005 | 0.005 | 94.50 | 0.15 |
| 4 | $A = U_r\Sigma_r$ | 0.001 | 0.0005 | 94.50 | 0.14 |
| 4 | $A = U_r\Sigma_r$ | 0.001 | 0.001 | 94.38 | 0.16 |
| 4 | $A = U_r\Sigma_r$ | 0.001 | 0.005 | 94.21 | 0.10 |
| 4 | $A = U_r\Sigma_r$ | 0.005 | 0.0005 | 91.86 | 20.21 |
| 4 | $A = U_r\Sigma_r$ | 0.005 | 0.001 | 91.28 | 16.27 |
| 4 | $A = U_r\Sigma_r$ | 0.005 | 0.005 | 50.92 | 19.47 |
| 4 | $A = U_r$ | 0.0005 | 0.0005 | 93.81 | 0.05 |
| 4 | $A = U_r$ | 0.0005 | 0.001 | 93.69 | 0.16 |
| 4 | $A = U_r$ | 0.0005 | 0.005 | 93.92 | 0.14 |
| 4 | $A = U_r$ | 0.001 | 0.0005 | 93.69 | 0.20 |
| 4 | $A = U_r$ | 0.001 | 0.001 | 94.04 | 0.16 |
| 4 | $A = U_r$ | 0.001 | 0.005 | 93.92 | 0.19 |
| 4 | $A = U_r$ | 0.005 | 0.0005 | 94.38 | 0.32 |
| 4 | $A = U_r$ | 0.005 | 0.001 | 94.15 | 0.34 |
| 4 | $A = U_r$ | 0.005 | 0.005 | 94.27 | 0.24 |

Table 20: Comparison of LoRA-XS initialization with and without singular values on SST-2.

| Rank | Init Type | LR | CLS LR | Median Score | Std Dev |
|---|---|---|---|---|---|
| 4 | $A = U_r\Sigma_r$ | 0.0005 | 0.0005 | 86.27 | 0.63 |
| 4 | $A = U_r\Sigma_r$ | 0.0005 | 0.001 | 85.54 | 1.01 |
| 4 | $A = U_r\Sigma_r$ | 0.0005 | 0.005 | 85.54 | 1.23 |
| 4 | $A = U_r\Sigma_r$ | 0.001 | 0.0005 | 86.52 | 0.73 |
| 4 | $A = U_r\Sigma_r$ | 0.001 | 0.001 | 86.76 | 1.11 |
| 4 | $A = U_r\Sigma_r$ | 0.001 | 0.005 | 86.76 | 0.81 |
| 4 | $A = U_r\Sigma_r$ | 0.005 | 0.0005 | 69.85 | 6.25 |
| 4 | $A = U_r\Sigma_r$ | 0.005 | 0.001 | 74.26 | 6.69 |
| 4 | $A = U_r\Sigma_r$ | 0.005 | 0.005 | 68.63 | 1.79 |
| 4 | $A = U_r$ | 0.0005 | 0.0005 | 82.60 | 0.75 |
| 4 | $A = U_r$ | 0.0005 | 0.001 | 83.33 | 0.67 |
| 4 | $A = U_r$ | 0.0005 | 0.005 | 82.11 | 0.50 |
| 4 | $A = U_r$ | 0.001 | 0.0005 | 86.03 | 0.57 |
| 4 | $A = U_r$ | 0.001 | 0.001 | 86.52 | 0.77 |
| 4 | $A = U_r$ | 0.001 | 0.005 | 85.78 | 0.25 |
| 4 | $A = U_r$ | 0.005 | 0.0005 | 87.50 | 0.50 |
| 4 | $A = U_r$ | 0.005 | 0.001 | **88.24** | 0.24 |
| 4 | $A = U_r$ | 0.005 | 0.005 | 87.50 | 1.27 |

Table 21: Comparison of LoRA-XS initialization with and without singular values on MRPC.

| Rank | Init Type | LR | CLS LR | Median Score | Std Dev |
|---|---|---|---|---|---|
| 4 | $A = U_r\Sigma_r$ | 0.0005 | 0.0005 | 55.01 | 0.58 |
| 4 | $A = U_r\Sigma_r$ | 0.0005 | 0.001 | 55.47 | 0.63 |
| 4 | $A = U_r\Sigma_r$ | 0.0005 | 0.005 | 58.63 | 1.21 |
| 4 | $A = U_r\Sigma_r$ | 0.001 | 0.0005 | 58.04 | 1.31 |
| 4 | $A = U_r\Sigma_r$ | 0.001 | 0.001 | 57.46 | 0.75 |
| 4 | $A = U_r\Sigma_r$ | 0.001 | 0.005 | **60.29** | 1.06 |
| 4 | $A = U_r\Sigma_r$ | 0.005 | 0.0005 | 43.78 | 14.36 |
| 4 | $A = U_r\Sigma_r$ | 0.005 | 0.001 | 47.60 | 3.72 |
| 4 | $A = U_r\Sigma_r$ | 0.005 | 0.005 | 25.51 | 15.58 |
| 4 | $A = U_r$ | 0.0005 | 0.0005 | 51.21 | 0.67 |
| 4 | $A = U_r$ | 0.0005 | 0.001 | 52.07 | 0.90 |
| 4 | $A = U_r$ | 0.0005 | 0.005 | 54.06 | 0.63 |
| 4 | $A = U_r$ | 0.001 | 0.0005 | 52.59 | 0.33 |
| 4 | $A = U_r$ | 0.001 | 0.001 | 52.40 | 0.65 |
| 4 | $A = U_r$ | 0.001 | 0.005 | 55.52 | 0.73 |
| 4 | $A = U_r$ | 0.005 | 0.0005 | 56.79 | 1.08 |
| 4 | $A = U_r$ | 0.005 | 0.001 | 57.22 | 0.52 |
| 4 | $A = U_r$ | 0.005 | 0.005 | 58.74 | 0.63 |

Table 22: Comparison of LoRA-XS initialization with and without singular values on CoLA.

| Rank | Init Type | LR | CLS LR | Median Score | Std Dev |
|------|-----------|-----|--------|--------------|---------|
| 4 | $A = U_r\Sigma_r$ | 0.0005 | 0.0005 | 90.88 | 0.15 |
| 4 | $A = U_r\Sigma_r$ | 0.0005 | 0.001 | 90.68 | 0.05 |
| 4 | $A = U_r\Sigma_r$ | 0.0005 | 0.005 | 90.28 | 0.20 |
| 4 | $A = U_r\Sigma_r$ | 0.001 | 0.0005 | **90.98** | 0.09 |
| 4 | $A = U_r\Sigma_r$ | 0.001 | 0.001 | 90.96 | 0.11 |
| 4 | $A = U_r\Sigma_r$ | 0.001 | 0.005 | 90.72 | 0.21 |
| 4 | $A = U_r\Sigma_r$ | 0.005 | 0.0005 | 50.54 | 0.75 |
| 4 | $A = U_r\Sigma_r$ | 0.005 | 0.001 | 50.54 | 0.00 |
| 4 | $A = U_r\Sigma_r$ | 0.005 | 0.005 | 50.54 | 0.00 |
| 4 | $A = U_r$ | 0.0005 | 0.0005 | 88.43 | 0.19 |
| 4 | $A = U_r$ | 0.0005 | 0.001 | 85.30 | 0.00 |
| 4 | $A = U_r$ | 0.001 | 0.0005 | 89.86 | 0.18 |
| 4 | $A = U_r$ | 0.001 | 0.001 | 89.80 | 0.21 |
| 4 | $A = U_r$ | 0.001 | 0.005 | 89.07 | 0.50 |
| 4 | $A = U_r$ | 0.005 | 0.0005 | 90.87 | 0.01 |
| 4 | $A = U_r$ | 0.005 | 0.001 | 90.92 | 0.15 |
| 4 | $A = U_r$ | 0.005 | 0.005 | 90.48 | 0.29 |

Table 23: Comparison of LoRA-XS initialization with and without singular values on QNLI.

