# OpenReview forum: "LoRA-XS: Low-Rank Adaptation with Extremely Small Number of Parameters"
_ICLR.cc/2025/Conference — ICLR 2025 Conference Withdrawn Submission_

### Official Review · Reviewer_1rMv · 2024-10-31

**Soundness:** 3
**Presentation:** 3
**Contribution:** 3
**Rating:** 5
**Confidence:** 4

**Summary:**

This work introduces LoRA-XS, a novel method for low-rank adaptation that significantly reduces the number of trainable parameters in large language models (LLMs) while maintaining or even improving performance.

**Strengths:**

1.   LoRA-XS inserts a small, trainable $r \times r$ weight matrix between frozen low-rank matrices constructed from the SVD of the original weight matrix, providing a lightweight mechanism for fine-tuning that requires much less storage space.
2. Despite the significant reduction in parameters, LoRA-XS shows superior or competitive performance across a variety of benchmarks, including GLUE, GSM8K, MATH, and eight common sense reasoning datasets, when compared to LoRA and other recent methods like VeRA.
3. The experiments reveal that self-attention layers can tolerate a high degree of dimensionality reduction, whereas output dense layers benefit from retaining a larger portion of the singular spectrum. This suggests that LoRA-XS offers flexibility in terms of how many and which singular vectors to retain, depending on the specific requirements of the task and the model architecture.
4. The ablation studies provide valuable insights into the role of singular vectors within transformer weights, showing that top singular vectors retain the most task-relevant knowledge, while middle and bottom vectors contribute less to task performance. This finding supports the design choice of using top singular vectors in LoRA-XS.

**Weaknesses:**

Q1. The performance of LoRA-XS is highly dependent on which singular vectors (top, middle, or bottom) are retained for each module. For example, the self-attention layers may perform well even when only a small fraction of the top singular values is kept, but the output dense layers might require a larger portion of the singular spectrum to maintain good performance. This sensitivity suggests that careful tuning is necessary for optimal results.

Q2. LoRA-XS relies on using SVD to construct the low-rank matrices and then inserting a trainable  $r \times r$ matrix. How to initialize this small matrix and how to choose the rank $r$?

Q3. What is the time overhead of the initial computation of the SVD for the weight matrices of the base model, particularly for very large models?

**Questions:**

Please see weaknesses.

---

> ### Author Response · Authors · 2024-11-14
> **Response to Reviewer 1rMv**
>
> ***Q1:** “The performance of LoRA-XS is highly dependent on which singular vectors (top, middle, or bottom) are retained for each module. For example, the self-attention layers may perform well even when only a small fraction of the top singular values is kept, but the output dense layers might require a larger portion of the singular spectrum to maintain good performance. This sensitivity suggests that careful tuning is necessary for optimal results.”*
>
> - Thank you for raising the point regarding the sensitivity of LoRA-XS to the singular vectors retained for different layers. We agree that the performance of LoRA-XS is influenced by the specific choice of singular vectors, and we will highlight this potential for improvement in the revised version of the paper.
>
> ***Q2:** “LoRA-XS relies on using SVD to construct the low-rank matrices and then inserting a trainable matrix. How to initialize this small matrix and how to choose the rank?”*
>
> - To clarify, we initialize the r×r matrix using a Gaussian distribution N(0,σ^2), where σ is set to a small value (specifically σ=10^(−5)) for all our LoRA-XS experiments. Similar to LoRA, this initialization ensures that we essentially start the fine-tuning from the pre-trained model weights.
>
> - Regarding the choice of r, LoRA-XS is designed to be flexible, allowing users to select the rank based on available memory and computational resources. We agree that dynamic rank selection is an intriguing idea that could be explored to optimize performance further, depending on the task or model size. This is an area for future research, and we will include this potential direction in the updated paper.
>
> ***Q3:** “What is the time overhead of the initial computation of the SVD for the weight matrices of the base model, particularly for very large models?”*
>
> - Thank you for highlighting the computational cost of SVD. To address this, we note that SVD is a one-time computation for each model weight matrix, which is considerably faster than full fine-tuning and is offset by LoRA-XS's efficiency gains during adaptation.
>
> - To quantify this, we conducted experiments on an H100 GPU using RoBERTa-large across 24 layers. Specifically, we compared the total fine-tuning time with the SVD initialization time for all matrices on the SST-2 (20 epochs) and MRPC (50 epochs) tasks with ranks of 4 and 25.
>
> The results are as follows:
> | Task | rank |  TFT | SVDT | SVDT/TFT (%) |
> |:----:|:----:|:----:|:----:|:------------:|
> | SST2 |   4  | 7310 | 10.6 |     0.14%    |
> | SST2 |  25  | 6980 | 19.1 |     0.27%    |
> | MRPC |   4  | 1215 | 10.8 |     0.89%    |
> | MRPC |  25  | 3104 | 18.9 |     0.61%    |
>
> Notation:
>
> Total Fine-Tuning Time (in seconds) – TF_T
>
> SVD Initialization for all adapters (in seconds) – SVD_T
>
> - These results demonstrate that SVD costs are minimal relative to the overall fine-tuning time, confirming that the initialization step does not significantly impact computational efficiency. We will include these findings in our paper for further clarity.

---

### Official Review · Reviewer_1J8q · 2024-11-01

**Soundness:** 2
**Presentation:** 3
**Contribution:** 2
**Rating:** 3
**Confidence:** 5

**Summary:**

The pape presents a novel method, LoRA-XS, for parameter-efficient fine-tuning of LLMs. This method is a successor to LoRA, which is popular for reducing the number of trainable parameters during fine-tuning.

The key contributions of LoRA-XS are:

- Extreme Parameter Efficiency: LoRA-XS introduces a small, trainable r × r matrix between frozen low-rank matrices constructed using Singular Value Decomposition of the original weight matrix. This setup significantly reduces the number of trainable parameters compared to LoRA, allowing over 100x reduction in models like GPT-3, making it possible to serve millions of personalized models with minimal memory requirements.

- Performance: The evaluations across multiple benchmarks, such as GLUE, GSM8K, MATH, and eight commonsense reasoning datasets, show that LoRA-XS performs competitively or better than LoRA and recent methods like VeRA, while being significantly more parameter-efficient.

- Independence from Model Dimensions: Unlike existing methods, the number of trainable parameters in LoRA-XS is independent of model dimensions, offering greater flexibility and storage efficiency, especially for large-scale models.

- Ablation Study: The paper also conducts an ablation study on the importance of singular vectors in transformer weights, providing insights into the mechanism that drives LoRA-XS's enhanced efficiency.

Overall, LoRA-XS provides a powerful, storage-efficient approach for scaling and personalizing large language models, maintaining competitive performance with drastically fewer trainable parameters.

**Strengths:**

**Originality**: The originality of the paper lies in its novel approach to parameter-efficient fine-tuning for LLMs. LoRA-XS builds upon the foundation of LoRA by incorporating a small, trainable matrix inserted between frozen low-rank matrices derived from Singular Value Decomposition. This innovative strategy introduces an extreme reduction in trainable parameters compared to prior methods like LoRA and VeRA. By making the number of parameters independent of model dimensions, the authors have effectively eliminated a significant limitation of existing parameter-efficient fine-tuning techniques.

**Quality**:The paper is supported by rigorous theoretical derivations and comprehensive empirical evaluations. The authors provide a well-founded theoretical explanation of why LoRA-XS achieves its extreme parameter efficiency by deriving optimal parameter subspaces using truncated SVD. The empirical results on diverse benchmarks, including GLUE, GSM8K, MATH, and commonsense reasoning datasets, demonstrate that LoRA-XS achieves competitive or superior performance to both LoRA and VeRA, despite using significantly fewer parameters.

**Clarity**:The paper is generally well-written and structured, making the proposed method accessible to a broad audience. The use of visual aids, such as diagrams comparing LoRA and LoRA-XS, helps in understanding the differences between these methods.

**Significance**:LoRA-XS attempts to address the significant storage and memory challenges associated with serving millions of personalized models by reducing trainable parameters by over 100x compared to LoRA.

**Weaknesses:**

There are some weaknesses that impact the overall significance of the work:

1. **Limited Applicability of Memory Savings**: The authors argue that the 100x memory saving benefits the inference of millions of adapters. However, even the most popular model, such as LLaMA3-8B, only has 494 adapters available on Hugging Face ([source](https://huggingface.co/models?other=base_model:adapter:meta-llama/Meta-Llama-3-8B)). This limited usage weakens the practical significance of the proposed memory savings, as the demand for such an extensive number of adapters is currently not evident.

2. **Computation Overhead**: LoRA-XS introduces an additional matrix multiplication per LoRA module, which results in computational overhead during both training and inference. This represents a tradeoff between memory efficiency and computational cost. Unfortunately, the paper does not provide a detailed system footprint analysis to demonstrate the practical tradeoff between memory savings and computational overhead, which is crucial for understanding the overall benefit of the approach.

3. **Inconsistent Performance Across Tasks**: The experimental results show that LoRA-XS does not consistently outperform or match the base models across different downstream tasks. For example, the LLaMA2-7B base model achieves a performance of 77.2 on HellaSwag according to the LLaMA2 paper ([source](https://arxiv.org/pdf/2307.09288)), whereas LoRA-XS achieves a lower score of 75.4 after fine-tuning, indicating negative optimization. Similarly, the performance on BoolQ for the LLaMA2-7B base model is 77.4, but drops to 67.2 after fine-tuning with LoRA-XS. Additionally, ARC-challenge for the LLaMA3-8B base model achieves 78.6, while LoRA-XS results in 76.5, according to the LLaMA3 blog ([source](https://ai.meta.com/blog/meta-llama-3/)). These inconsistencies highlight that LoRA-XS may not provide consistent benefits across all tasks, which reduces its overall reliability and applicability.

4. **Lack of Guidance for Hyperparameter Selection**: The paper lacks advice or experimental evidence on how to choose an appropriate value for the hyperparameter r of the small trainable matrix R. This omission leaves practitioners without clear guidelines for selecting the best configuration for different tasks or models, which could impact the ease of adoption and effectiveness of LoRA-XS.

**Questions:**

There are some questions for this paper:

1. Gemma 7B base model's performance on MATH is 24.3 according to Hugging Face ([source](https://huggingface.co/google/gemma-7b)). Why is the Full FT performance of 22.74 even lower than the base model?

2. In Table 3, LoRA-XS underperforms compared to LoRA in MATH task. Have you tried using larger ranks for LoRA-XS to improve performance?

3. In Appendix C.2, you train for 2 epochs on MetaMathQA for both LoRA and LoRA-XS, while the typical fine-tuning epoch number is usually 3. How did you choose this hyperparameter, and is it optimal for LoRA or LoRA-XS?

4. In Appendix D, you find that the top subsets of singular vectors in RoBERTa-large retain the most task-relevant knowledge on MRPC, SST2, and MNLI. Does this result hold for different models (e.g., larger models like LLaMA3-70B) and other task scenarios (such as coding and math)? Could you provide some theoretical insights or analysis?

---

> ### Author Response · Authors · 2024-11-14
> **Response to Reviewer 1J8q**
>
> ***W1:** Limited Applicability of Memory Savings*
>
> - Thank you for pointing out the practical scope of LoRA-XS's memory savings. To clarify, the primary advantage of LoRA-XS is not limited to accommodating millions of adapters within a single model instance but rather to efficiently storing millions of personalized model variants. We envision a future of highly personalized AI, where LoRA-XS could store minimal r×r matrices for each personalized model, ideal for applications in education, healthcare, or other settings where per-user or per-task models may be beneficial. We will clarify this use case in the final version.
>
> ***W2:** Computation Overhead*
>
> - We appreciate your question regarding computational overhead. To address this, LoRA-XS does not introduce additional inference overhead as it merges directly into model weights, similar to LoRA.
>
> - Regarding training time: to demonstrate the minimal effect the extra multiplication has on LoRA-XS training time, we trained RoBERTa-large using both methods on the MRPC task for 10 epochs, for three different ranks on an H100 GPU. As you can see in the following table, the runtime is minimally affected as the extra multiplication is performed over the latent dimension.
>
> |  model  | rank | task |  Runtime (s)  |
> |:-------:|:----:|:----:|:-------------:|
> |   LoRA  |  16  | MRPC |      198.8    |
> | LoRA-XS |  16  | MRPC |      200.7    |
> |   LoRA  |  32  | MRPC |      200.6    |
> | LoRA-XS |  32  | MRPC |      201.7    |
> |   LoRA  |  64  | MRPC |      204.5    |
> | LoRA-XS |  64  | MRPC |      204.8    |
>
>
> ***W3:** Inconsistent Performance Across Tasks*
>
> - Thank you for discussing LoRA-XS performance across tasks. LoRA-XS is designed for resource efficiency, making it particularly useful in settings where storage and memory limitations are a priority. Due to computational constraints, we performed a limited hyperparameter search for mathematical reasoning experiments on the 7B models, with most parameters set according to prior works. We believe additional tuning would close the performance gap with LoRA, as seen in tasks like GLUE, GSM8K, and commonsense reasoning, where LoRA-XS consistently performs competitively while being more parameter-efficient. We will further elaborate on this point in the revised paper.
>
> ***W4:** Lack of Guidance for Hyperparameter Selection*
>
> - We recognize the importance of guidance on selecting an optimal r value. We see this as a key topic for future research, as optimal rank selection likely depends on both task complexity and model size. LoRA-XS’s strength lies in its flexibility, allowing r to be selected to meet specific memory constraints. We see the rank selection as a promising direction for future research, and we will address this in the paper.
>
> ***Q1:** “Gemma 7B base model's performance on MATH is 24.3 according to Hugging Face (source). Why is the Full FT performance of 22.74 even lower than the base model?”*
>
> - Thank you for this question. The number we reported was taken from previous work, and we conducted our experiments using a 0-shot prompting setting. The scores reported on Hugging Face, however, are based on a 4-shot setting. We will clarify this in the paper.
>
> ***Q2:** “In Table 3, LoRA-XS underperforms compared to LoRA in the MATH task. Have you tried using larger ranks for LoRA-XS to improve performance?”*
>
> - Due to computational limitations, we tested LoRA-XS with a single hyperparameter configuration. However, we believe that a more extensive hyperparameter search would help narrow the performance gap.
>
> ***Q3:** “In Appendix C.2, you train for two epochs on MetaMathQA for both LoRA and LoRA-XS, while the typical fine-tuning epoch number is usually 3. How did you choose this hyperparameter, and is it optimal for LoRA or LoRA-XS?”*
>
> - Thank you for raising this point. In order to have a fair comparison with the baseline, we followed the same number of epochs as those used for the baseline training. We will clarify this in the revised paper.
>
> ***Q4:** “In Appendix D, you find that the top subsets of singular vectors in RoBERTa-large retain the most task-relevant knowledge on MRPC, SST2, and MNLI. Does this result hold for different models (e.g., larger models like LLaMA3-70B) and other task scenarios (such as coding and math)? Could you provide some theoretical insights or analysis?”*
>
> - Due to computational constraints, we focused our ablation study on RoBERTa-large. However, the strong performance results of LoRA-XS across diverse, larger-scale benchmarks in the main paper indicate that our method is indeed generalizable to even larger models.

---

> > ### Comment · Reviewer_1J8q · 2024-11-27
> > **Response to Authors**
> >
> > Thank you for your detailed responses and clarifications. I appreciate the additional context you provided regarding the applicability of LoRA-XS. But I still noticed that the communication overhead results appear to be inconsistent when compared to previous work, such as the results reported in the dLoRA paper. And in current systems, disk storage is often quite affordable. In addition, the finetuning results you provided reveal that, in some cases, LoRA-XS underperforms even when compared to the base model. Given these concerns, I am inclined to maintain my original score.

---

### Official Review · Reviewer_KVyz · 2024-11-01

**Soundness:** 3
**Presentation:** 3
**Contribution:** 2
**Rating:** 5
**Confidence:** 3

**Summary:**

The paper introduces LoRA-XS, an efficient variant of LoRA aimed at reducing the number of trainable parameters in large language models (LLMs) without compromising the performance. LoRA-XS initializes and freezes the standard low-rank matrices (AB) as top-$r$ singular vectors  derived from SVD of pre-trained weight matrices $W$ and adds trainable $r$ x $r$ matrix between them for flexibility, which is completely independent of the model dimensions. Their approach improves the parameter efficiency by constraining the parameter space for LoRA adaptation from $2nr$ to $r^{2}$, maintaining the model performance in overall. The paper also provides the theoretical derivation that LoRA-XS basically implements an orthogonal projection of any gradient update of $W$ onto a low-rank subspace spanned by the top singular vectors of $W$. The benefit of using top singular basis of pre-trained weights compared to middle and bottom basis or simply using random initialization is empirically demonstrated by extensive ablation studies.

**Strengths:**

Unlike AdaLoRA that uses SVD to dynamically adjust the rank of each adaptation matrices, LoRA-XS simply uses the top-r singular vectors  from SVD of pre-trained weights to construct learnable $r$ x $r$ matrices. The approach is quite simple and straightforward yet accomplishes the following contributions.

1. Improve parameter efficiency while maintaining the model performance

Unlike standard LoRA, where trainable parameters scale with the model's hidden dimension, LoRA-XS’s adaptation matrix is $r$ × $r$, meaning the trainable parameter count is fixed at $r^{2}$, independent of the model’s width. This design allows efficient fine-tuning of large language models with minimal additional memory. This $r$ × $r$ matrix effectively captures task-specific adjustments in the weight space, leveraging the important information encoded in top-singular basis and maintains competitive performance with fewer parameters.

2. Extensive ablation studies to empirically justify the design rationale

The paper includes comprehensive ablation studies to validate the importance of using top-r singular vectors. They compare the model performance with varying subspaces (top, middle, bottom singular vectors), rank fractions, and initialization methods to understand the impact of basis choice on model performance. Results consistently show that the top-r singular vectors retain the most task-relevant information (except intermediate.dense layer, which seems heavily affected by the input data), aligning well with the directions already present in the pre-trained weights. This finding demonstrates that LoRA-XS can maintain competitive performance by focusing only on these crucial components, while significantly reducing parameter usage.

**Weaknesses:**

1. Computational cost for SVD for each pre-trained weight matrix.

Since LoRA-XS freezes the orthogonal matrices for pre-trained $W$, they need to fully compute the SVD instead of approximating them via regularization techniques as AdaLoRA did. The computational load for computing SVD for every and each $W$ would be very heavy as the complexity for SVD is $O(min(d_1, d_2)d_1d_2)$. My concern is that the initial cost from computing SVD might offset the parameter efficiency gains during the adaptation phase, especially for LLMs with large model width.

2. Ambiguous theoretical role of the $r$ x  $r$ matrix.

Besides of adding flexibility to frozen orthogonal top-r singular vectors, what's the theoretical role of $r$ x  $r$ matrix? The paper briefly mentions about the slight shift in gradient distribution, but can't see any clear explanation about the theoretical relation between this $r$ x  $r$ matrix and distribution shift to adjust to the fine-tuning data.

3. A minor concern : The approach lacks novelty.

The simplicity of LoRA-XS can be seen as both a strength and a limitation. On one hand, the approach lacks conceptual novelty, as other LoRA-based methods have similarly employed SVD to capture core singular components from pre-trained weight matrices. Simply extracting the top-r singular vectors and freezing them with additional $r$ x $r$ matrix for task-specific tuning seems too heuristic. However, LoRA-XS still provides strong empirical evidences to justify its choice of basis, demonstrating that focusing on the top singular components can achieve competitive performance with high parameter efficiency.

**Questions:**

Computational Cost for SVD: Have you considered alternative techniques to reduce the computational burden of performing full SVD on each weight matrix?

Theoretical Role of $r$ x $r$ Matrix: Can you clarify the theoretical role of the $r$ x $r$ matrix in shifting the weight distribution to adapt to fine-tuning data?

---

> ### Author Response · Authors · 2024-11-14
> **Response to Reviewer KVyz**
>
> ***W1 & Q1:** “Computational cost for SVD for each pre-trained weight matrix.”*
>
> - Thank you for highlighting the computational cost of SVD. To address this, we note that SVD is a one-time computation for each model weight matrix, which is considerably faster than full fine-tuning and is offset by LoRA-XS's efficiency gains during adaptation.
>
> - To quantify this, we conducted experiments on an H100 GPU using RoBERTa-large across 24 layers. Specifically, we compared the total fine-tuning time with the SVD initialization time for all matrices on the SST-2 (20 epochs) and MRPC (50 epochs) tasks with ranks of 4 and 25.
>
> The results are as follows:
> | Task | rank |  TFT | SVDT | SVDT/TFT (%) |
> |:----:|:----:|:----:|:----:|:------------:|
> | SST2 |   4  | 7310 | 10.6 |     0.14%    |
> | SST2 |  25  | 6980 | 19.1 |     0.27%    |
> | MRPC |   4  | 1215 | 10.8 |     0.89%    |
> | MRPC |  25  | 3104 | 18.9 |     0.61%    |
>
> Notation:
>
> Total Fine-Tuning Time (in seconds) – TF_T
>
> SVD Initialization for all adapters (in seconds) – SVD_T
>
> - These results demonstrate that SVD costs are minimal relative to the overall fine-tuning time, confirming that the initialization step does not significantly impact computational efficiency. We will include these findings in our paper for further clarity.
>
> ***W2 & Q2:** “Ambiguous theoretical role of the r×r matrix.”*
>
> - Thank you for prompting us to elaborate on the theoretical role of the r×r matrix in LoRA-XS. The r×r matrix (initialized with Gaussian noise) serves as a task-adaptive layer. By fine-tuning this matrix, LoRA-XS effectively performs latent editing within a constrained r-dimensional subspace, driven by task-relevant components. We will expand on this in the paper to provide more insight into the functional role of this matrix.
>
> ***W3:** “A minor concern : The approach lacks novelty.”*
>
> - We appreciate your perspective on LoRA-XS’s simplicity, which we see as a strength. This approach offers both ease of implementation and practical utility. Looking forward, we anticipate growing demand for personalized models, which would enhance LoRA-XS’s usability even further. By storing only the small r×r matrices and computing the fixed SVD later as needed, LoRA-XS allows efficient storage and scalability.

---

> > ### Comment · Reviewer_KVyz · 2024-11-25
> >
> > I appreciate the response from the authors and I'd like to raise another concern regarding the limited capacity of LoRA-XS compared to standard LoRA, as highlighted in Table 2. There are some noticeable trade-offs in accuracy for certain settings (LLaMA2-7B on commonsense reasoning tasks), suggesting the potential limitations in its representational power relative to LoRA. Specifically, LoRA-XS shows a significant drop in performance from 83.6 to 75.4 and 81.0 to 74.6 on HellaSwag and QBQA, respectively. Could you provide the performance of base model (no fine-tuning) under the same experimental settings as Table 2? This would give a clearer comparison of the gains achieved by each fine-tuning method, LoRA and LoRA-XS.

---

### Official Review · Reviewer_XSC7 · 2024-11-03

**Soundness:** 3
**Presentation:** 3
**Contribution:** 2
**Rating:** 5
**Confidence:** 3

**Summary:**

The paper proposes a LoRA-XS method for LLM PEFT with extremely small trainable parameters. The authors evaluate various downstream tasks to show the good performance compared to other baselines.

**Strengths:**

1. The authors propose LoRA-XS, which uses only extremely small trainable parameters for PEFT scenarios.

**Weaknesses:**

1. The proposed method aims to learn the tra
2. The results of full parameter fine-tuning for Mixtral and Gemma are directly taken from other resources, and the performances are lower than LoRA FT, which is a little bit strange.
3. LoRA fine-tuning results for LlaMA series are also directly taken from other resources.
4. LLM with LoRA-XS requires more space to store U, V, and Σr, and requires one more matrix multiplication compared to LoRA.

**Questions:**

I am concerned about the evaluation part. Directly taking the results from other papers is not the wrong choice, but it could be better to evaluate all baselines by the authors themselves so that the evaluation could be fair enough by running on the same platform and controlling all hyper-parameters. It is because different platforms (e.g., different CUDA versions, different torch, transformers versions, AMP training, whether using BF16, etc) may lead to different results.

1. I know the full parameter fine-tuning may rely on too much computational resource, could the authors do their best efforts to do LoRA fine-tuning on LlaMA 2-7B and LlaMA3-8B evaluation?
2. Directly doing SVD on weight matrices usually shows unignorable errors; could the author explain why the proposed method (i.e., applying a weight W) could reduce this error?
3. Could the author do the evaluation for efficiency in terms of both HBM costs and the computational consumption by applying one more Matmul?

---

> ### Author Response · Authors · 2024-11-14
> **Response to Reviewer XSC7**
>
> ***W1:***
> - Could you please clarify what you meant by "learn the tra"? We want to address your point accurately.
>
> ***W2 & W3:**
> “The results of full parameter fine-tuning for Mistral (7B) and Gemma (7B) are directly taken from other resources, and the performances are lower than LoRA FT, which is a little bit strange.”
> “LoRA fine-tuning results for LLaMA series are also directly taken from other resources.”*
>
> - For this experiment, we indeed used the same script as the prior work to train our LoRA-XS models to ensure a fair comparison with LoRA and full fine-tuning baselines. We acknowledge the point about the possible variations in platform conditions when using results from prior resources. However, due to resource constraints, similarly to other works, we’ve used the results from previous reliable studies. We will ensure that all sources are explicitly cited in the final version to clarify this.
>
> ***W4:** “LLM with LoRA-XS requires more space to store U, V, and Σr, and requires one more matrix multiplication compared to LoRA.”*
>
> - Thank you for raising this question. In LoRA-XS, only the small trainable r×r matrices are fine-tuned, while U and V remain fixed during fine-tuning. Furthermore, as LoRA-XS merges these trainable weights directly into the model during inference, there’s no need to retain additional matrices after fine-tuning (no added computational overhead during inference).
>
> - If storing the adapters would be needed, we can store only the r×r matrices, as U and V, derived from SVD, remain constant and don’t require separate storage for each task or personalized model.
>
> ***Q1:** “I know the full parameter fine-tuning may rely on too much computational resource, could the authors do their best efforts to do LoRA fine-tuning on LlaMA 2-7B and LlaMA3-8B evaluation?”*
>
> - Due to computing limitations, we aimed to ensure a fair comparison by directly applying LoRA-XS modules for LlaMA 2-7B and LlaMA3-8B into the existing code-base from previous work (DoRA paper). We will make sure to mention this in the paper.
>
> ***Q2:** “Directly doing SVD on weight matrices usually shows unignorable errors; could the author explain why the proposed method (i.e., applying a weight W) could reduce this error?”*
>
> - Thank you for this question. While it’s true that direct SVD on weight matrices can introduce approximation errors, LoRA-XS is designed to mitigate this. By applying a small trainable matrix R between the fixed singular matrices U and V, LoRA-XS provides an adaptation that compensates for any discrepancies introduced by the SVD approximation.
>
> ***Q3:** “Could the author do the evaluation for efficiency in terms of both HBM costs and the computational consumption by applying one more Matmul?”*
>
> - We appreciate your question. LoRA-XS does require an additional matrix multiplication for R, but as this multiplication is done over the latent space, it’s computationally cheap and thus minimally affecting the model’s forward pass time. Moreover, the fixed U and V matrices are non-trainable, so they do not increase memory usage for gradients or optimizer states. At inference, we can merge R directly into the model weights, eliminating any extra computational or memory overhead. We will clarify this in the paper.
>
> - To demonstrate the minimal effect the extra multiplication has on LoRA-XS training time, we trained RoBERTa-large using both methods on the MRPC task for 10 epochs, for three different ranks on an H100 GPU. As you can see in the following table, the runtime is minimally affected as the extra multiplication is performed over the latent dimension.
>
> |  model  | rank | task |  Runtime (s)  |
> |:-------:|:----:|:----:|:-------------:|
> |   LoRA  |  16  | MRPC |      198.8    |
> | LoRA-XS |  16  | MRPC |      200.7    |
> |   LoRA  |  32  | MRPC |      200.6    |
> | LoRA-XS |  32  | MRPC |      201.7    |
> |   LoRA  |  64  | MRPC |      204.5    |
> | LoRA-XS |  64  | MRPC |      204.8    |

---

> > ### Comment · Reviewer_XSC7 · 2024-11-25
> >
> > I have read the rebuttal and willing to keep my score.
> >
> > The reason is that fine-tuning a model requires lots of hyper-parameters and relies on the version of CUDA/Torch, etc. Thus, a fair comparison should be made in evaluation.
> >
> > However, if the chair feels OK to directly reuse the results from the other papers, I will not hold on my point.

---

### Official Review · Reviewer_gY8e · 2024-11-04

**Soundness:** 3
**Presentation:** 3
**Contribution:** 2
**Rating:** 5
**Confidence:** 3

**Summary:**

This paper presents LoRA-XS (Low-Rank Adaptation with eXtremely Small number of parameters), a novel low-rank adaptation method for large language models. As the growth of large language models highlights the need for parameter-efficient fine-tuning methods, LoRA-XS is introduced to address the storage challenges of serving multiple LoRA modules. By inserting a small trainable weight matrix between frozen low-rank matrices constructed via Singular Value Decomposition (SVD), LoRA-XS significantly reduces trainable parameters while showing competitive or better performance. Evaluations across various benchmarks demonstrate its effectiveness and an ablation study provides insights into its underlying mechanisms. LoRA-XS offers a more efficient path for model personalization and task-specific optimization.

**Strengths:**

* LoRA-XS reduces the number of trainable parameters by over 100x in large-scale models without sacrificing performance, enabling the deployment of millions of personalized models with minimal memory overhead.
* LoRA-XS allows for precise control of the number of additional parameters and is independent of model dimensions, providing flexibility in memory usage and being more storage-friendly and adaptable.
* LoRA-XS outperforms LoRA and other recent methods like VeRA across various model sizes and a wide range of tasks while retaining the advantages of LoRA such as no architectural modifications and no additional inference latency.

**Weaknesses:**

* The discussion on related works such as including [1-2] can be further improved, given the rapid progress on LoRA-based parameter efficient fine-tuning of LLMs.
* The representation ability of LoRA-XS seems to be weaker than LoRA since the space resulting from LoRA-XS is much smaller than LoRA regarding the dimension. Could this make LoRA-XS easier to overfit than LoRA? Besides, is it possible for LoRA-XS to be harder to learn on more difficult tasks than LoRA?
* Cound the authors showcase the results on LLMs larger than 7B to further demonstrate the effectiveness of the LoRA-XS.
* The proposed Theorem 3.1 is rather general and only related to the derivation of LoRA-XS. Therefore, it feels a little overclaimed by writing it as a Theorem since the theoretical contributions are limited.

[1] QA-LoRA: Quantization-Aware Low-Rank Adaptation of Large Language Models, ICLR 2024

[2] Parameter Efficient Fine-Tuning with Discrete Fourier Transform, ICML 2024

**Questions:**

Please kindly refer to the Weaknesses.

---

> ### Author Response · Authors · 2024-11-14
> **Response to Reviewer gY8e**
>
> ***W1**: "The discussion on related works such as including [1-2] can be further improved, given the rapid progress on LoRA-based parameter efficient fine-tuning of LLMs."*
>
> - We agree that recent studies in parameter-efficient fine-tuning are relevant to this work. Thank you for suggesting additional references. We will incorporate [1] and [2] in our related work section and also expand our discussion on LoRA-based fine-tuning methods.
>
> ***W2**: “The representation ability of LoRA-XS seems to be weaker than LoRA since the space resulting from LoRA-XS is much smaller than LoRA regarding the dimension. Could this make LoRA-XS easier to overfit than LoRA? Besides, is it possible for LoRA-XS to be harder to learn on more difficult tasks than LoRA?”*
>
> - Thank you for discussing this point. We believe that the reduced parameter count in LoRA-XS could actually mitigate overfitting, as the model has fewer parameters to memorize the downstream tasks’ training examples. Our experiments also do not show evidence of an overfitting issue, as demonstrated across various models and tasks. We will add this clarification to the paper.
>
> ***W3**: “Could the authors showcase the results on LLMs larger than 7B to further demonstrate the effectiveness of the LoRA-XS.”*
>
> - We appreciate your suggestion to test on larger models. Due to resource constraints, following other similar works, we focused on multiple 7B and 8B models to demonstrate the generalization capabilities of our method.
>
> ***W4**: “The proposed Theorem 3.1 is rather general and only related to the derivation of LoRA-XS. Therefore, it feels a little overclaimed by writing it as a Theorem since the theoretical contributions are limited.”*
>
> - Thank you for your feedback on Theorem 3.1. We believe it provides insights that apply not only to LoRA-XS but also to other low-rank (parameter-efficient) adaptations. We will revise the phrasing to emphasize the theorem's relevance more accurately. Additionally, Theorem 3.1 offers valuable insights into the potential applicability of LoRA-XS (see Table 4 and Appendix F for more details). It helps explain why parameter-efficient adaptation of large language models (LLMs) can perform effectively for various fine-tuning tasks. If LoRA-XS shows lower performance on a specific task, it may indicate that this task diverges from the pre-training objective, suggesting that increasing the rank could improve adaptation. We will clarify this in the paper.

---

> > ### Comment · Reviewer_gY8e · 2024-11-25
> > **Thank you for the response**
> >
> > I would like to thank the authors for their replies. As certain of my concerns still stand, I would like to maintain my current rating.

---

### Official Review · Reviewer_QqvV · 2024-11-04

**Soundness:** 3
**Presentation:** 3
**Contribution:** 2
**Rating:** 3
**Confidence:** 4

**Summary:**

The paper proposes a extremely parameter-efficient low-rank adaptation methodology, called LoRA-XS. It uses Truncated SVD on the weight matrices of a pre-trained Transformer model to fix the adapter matrices A and B, and only learns a small $r \times r$ matrix between them. This allows the remaining modules to remain frozen, drastically reducing the number of learnable parameters while still achieving competitive performance compared to existing models. Experimental results show competitive performance on multiple datasets, including GLUE, GSM8K, and MATH.

**Strengths:**

- Experiments have been conducted on various domains, and sufficient ablation studies have been performed on the introduced modules.
- LoRA-XS can achieve similar performance than LoRA with significantly fewer trainable parameters, which greatly reduces learnable parameters.

**Weaknesses:**

W1. Since the input is mapped to a subspace of $W$ and adjusted in scale within that space, if the distribution of the dataset used for fine-tuning differs significantly from that of the pre-training dataset, it may not adapt adequately.

W2. In Table 3, although the computational efficiency is promising since the learnable parameters are drastically reduced, the performance decrease is noticeable. For instance, in the case of MATH dataset in Gemma model, the performance decreases from 31.28 when LoRA is used to 27.62 when LoRA-XS is applied. It is questionable whether this performance can be said to be comparable to the parameter efficiency advantage.

W3. In line 280, the authors refer to “(to $r$ eigenvalues).” However, in general, the pre-trained weights $W$ may not always be applicable for eigendecomposition, meaning that eigenvalues may not always exist.

(minor typo) I think in 311 line, $h=xW + x\Delta W=xW+xARB$ should be corrected as $h=Wx+\Delta Wx=Wx + ARBx$.

**Questions:**

Q1. The results reported in Table 1 exclude QQP and MNLI datasets from the GLUE task. As far as I know, these two datasets are significantly larger than other reported datasets. Is there any reason why you did not conduct experiments on these two datasets? Personally, I think it is because larger datasets are complex and require more ranks and parameters. Can you conduct additional experiments on these datasets?

Q2. Although the learnable parameters are greatly reduced, the matrix multiplication computation still seems similar to LoRA. I am wondering about the results for the actual runtime/GPU peak usage of the model.

---

> ### Author Response · Authors · 2024-11-14
> **Response to Reviewer QqvV**
>
> ***W1:** Since the input is mapped to a subspace of W and adjusted in scale within that space, if the distribution of the dataset used for fine-tuning differs significantly from that of the pre-training dataset, it may not adapt adequately.*
>
> - We evaluated LoRA-XS on diverse datasets, including GLUE, GSM8k, MATH, and eight commonsense reasoning tasks, and observed competitive or superior performance compared to LoRA, even on datasets that diverge from the pre-training distribution. For example, on SST-2 (a sentiment classification task), LoRA-XS initialized with random SVD performed only slightly better than when initialized with the SVD of pre-trained weights, indicating that it adapts well even when tasks differ somewhat from the pre-training objective (Table 4 and Appendix F).
>
> - In cases where LoRA-XS might underperform we can increase the rank to allow for a subspace with higher dimensionality. These experiments lie outside the scope of our current work but we agree it would be an interesting direction to explore dynamic rank allocation in the future.
>
> ***W2:** In Table 3, although the computational efficiency is promising since the learnable parameters are drastically reduced, the performance decrease is noticeable. For instance, in the case of MATH dataset in Gemma model, the performance decreases from 31.28 when LoRA is used to 27.62 when LoRA-XS is applied. It is questionable whether this performance can be said to be comparable to the parameter efficiency advantage.*
>
> - For our mathematical reasoning experiments, due to computational constraints we performed limited hyperparameter search for the 7B models and set most parameters according to prior works. We believe additional tuning would narrow the gap with LoRA, as seen in other tasks like GLUE, GSM8K, and commonsense reasoning, where LoRA-XS performs competitively while being more parameter-efficient.
>
> ***W3:** In line 280, the authors refer to “(to eigenvalues).” However, in general, the pre-trained weights may not always be applicable for eigendecomposition, meaning that eigenvalues may not always exist.*
>
> - Thank you for catching the terminology inconsistency (L280). We intended to refer to the first r singular values, not eigenvalues. We will correct this in the manuscript. Regarding implementation, we use randomized SVD to decompose the pre-trained weights.
>
> ***Q1:** The results reported in Table 1 exclude QQP and MNLI datasets from the GLUE task. As far as I know, these two datasets are significantly larger than other reported datasets. Is there any reason why you did not conduct experiments on these two datasets? Personally, I think it is because larger datasets are complex and require more ranks and parameters. Can you conduct additional experiments on these datasets?*
>
> - Due to resource limitations, we prioritized a representative subset of GLUE and did not include QQP and MNLI, which are larger and would require additional resources for extensive hyperparameter tuning. Recognizing the importance of these datasets, we will aim to add results for QQP and MNLI in the final paper.
>
> ***Q2:** “Although the learnable parameters are greatly reduced, the matrix multiplication computation still seems similar to LoRA. I am wondering about the results for the actual runtime/GPU peak usage of the model.”*
>
> - Regarding runtime: During inference, as we merge LoRA-XS parameters into the pre-trained weights, the inference will be similar to the base model (similar to LoRA). During training time, compared to LoRA, we do have an extra matrix multiplication, which is performed on the latent space, which makes it computationally cheap. To demonstrate the minimal effect the extra multiplication has on LoRA-XS training time, we trained RoBERTa-large using both methods on the MRPC task for 10 epochs, for three different ranks on an H100 GPU. As you can see in the following table, the runtime is minimally affected as the extra multiplication is performed over the latent dimension.
>
> |  model  | rank | task |  Runtime (s)  |
> |:-------:|:----:|:----:|:-------------:|
> |   LoRA  |  16  | MRPC |      198.8    |
> | LoRA-XS |  16  | MRPC |      200.7    |
> |   LoRA  |  32  | MRPC |      200.6    |
> | LoRA-XS |  32  | MRPC |      201.7    |
> |   LoRA  |  64  | MRPC |      204.5    |
> | LoRA-XS |  64  | MRPC |      204.8    |
>
> - Regarding fine-tuning memory usage: We performed all our experiments without quantizing the pre-trained model weights. This causes the memory footprint improvement for both LoRA and LoRA-XS negligible, although our method uses less memory. We expect to have a considerable improvement in memory footprint when training LoRA-XS with quantized weights, but we leave this to future work.
>
> We would like to point out that LoRA-XS’s application mainly lies in the scenario where we need to deploy multiple personalized models, and we can store them using significantly less memory (store only r x r matrices as SVD can be computed from pre-trained weights).

---

> > ### Comment · Reviewer_QqvV · 2024-11-25
> >
> > I deeply appreciate the thoughtful response from the authors. At this point, I intend to maintain my score.
> >
> > I have concerns regarding the performance degradation observed in large-scale models such as LLaMA-7B and Gemma, as well as the insufficiently verified performance on large datasets like MNLI and QQP in GLUE tasks.
> >
> > This issue seems closely related to the expressive capacity of LoRA. If the authors can clearly address this matter from a performance standpoint, I am willing to reconsider and increase my score.

---

### Note · Authors · 2024-12-10

I have read and agree with the venue's withdrawal policy on behalf of myself and my co-authors.